# Transient ice ring observed during the 15 January 2022 eruption of Hunga volcano

Andrew T. Prata [1,2] ✉, Roy G. Grainger [3], Isabelle A. Taylor[4] & Alyn Lambert [5]

The eruption of Hunga volcano on 15 January 2022 was an exceptional event in the satellite era. Record-breaking heights of the volcanic plume were reported, a large amount of water was injected into the stratosphere and a broad spectrum of atmospheric waves were detected. Here, we use satellite measurements to show that a transient ring of small ice particles (~2 μm) formed around the plume. We hypothesize that the ice ring was generated by the passage of an atmospheric wave triggered by a pressure pulse at the surface corresponding to a violent explosion that occurred during the 15 January 2022 eruption sequence. The passage of the atmospheric wave produced a transient rarefaction in the upper troposphere-lower stratosphere, which in turn led to oscillations in ambient temperature. Due to the supersaturated state of the atmosphere with respect to ice, ice particles formed in the wake of the radially propagating atmospheric wave, allowing an exceptional opportunity to study ice particle growth via vapour deposition. This atmospheric phenomenon serves as an important natural experiment that reveals the time scale on which ice particles nucleate and grow given an abrupt perturbation in ambient temperature.

Numerous atmospheric phenomena are associated with explosive volcanic eruptions. The observation of an umbrella cloud indicates that a volcanic plume has reached its level of neutral buoyancy[1–3]. Volcanic plumes with a warm centre relative to the surrounding umbrella can indicate that material has entered the stratosphere[4]. The presence of skirt and Pileus clouds indicates vigorous updrafts associated with plume dynamics where moist air in the atmosphere is pushed to the saturation point[5]. Mammatus clouds, sometimes formed on the underside of spreading umbrella plumes, can be an indication of ash-hydrometeor interactions and large-scale subsidence[6,7]. Wilson clouds (also known as condensation clouds) form due to the passage of a shockwave triggered by an explosion[8] and have been documented previously for volcanic eruptions[9], while abundant lightning detected by global networks often signals a high proportion of ice in volcanic plumes[3,10,11]. Recognising and understanding the formation mechanisms behind these atmospheric phenomena is important as they can provide information regarding the explosivity of the eruption, plume height, composition and microphysical processes within the plume.

On 13 January 2022 at around 15:37 UTC, substantial phreatomagmatic activity at Hunga volcano (Kingdom of Tonga) was observed by the Advanced Himawari Imager (AHI) aboard the Himawari-8 satellite. The 13 January eruption produced a relatively small amount of $SO_2$[0.06 Tg; 12] and generated an ice-rich plume, which formed an anvil that was maintained near (and above) the tropopause[13]. Eruptive activity continued through 14 January, which resulted in the formation of an ice cloud over the volcano by 15 January (see Supplementary Movie 1). On 15 January at around 04:07 UTC, Hunga volcano produced possibly the most energetic eruption since Krakatau in 1883[14,15]. Large amounts of $H_2O$ (~130–150 Tg) were injected into the stratosphere according to microwave limb sounder retrievals[16–19]. Initial estimates of $SO_2$ mass determined from ultraviolet vertical column density retrievals were modest[ ~0.4–0.5 Tg 12], but more recently hyperspectral thermal infrared measurements indicate a total mass of ~1 Tg $SO_2$ for the whole event[20]. The modest amounts of sulfur released by the Hunga eruption have recently been attributed to magma-seawater interactions[21]. Additionally, high concentrations of sea salts were found in volcanic ash collected in the first two weeks after the eruption[22].

By 04:37 UTC, the 15 January plume had rapidly established itself in the stratosphere, forming an umbrella cloud reminiscent of the Pinatubo umbrella[1] but with some notable differences. In the central portion of the umbrella, a dome-like structure formed with some regions of the plume entering the mesosphere and tendrils extending to 58 km[23,24]. From 04:47–05:17 UTC, after the dome collapsed, a grey diffuse cloud, surrounding the main eruption column, formed below the uppermost umbrella. The diffuse cloud had a striking circular shape, which can be appreciated by comparing the edge of the cloud with the 300 km circle radius

[1]Sub-Department of Atmospheric, Oceanic and Planetary Physics, Clarendon Laboratory, Oxford, UK. [2]Now at: CSIRO Environment, Clayton, Vic, Australia. [3]National Centre for Earth Observation, Atmospheric, Oceanic and Planetary Physics, University of Oxford, Oxford, UK. [4]COMET, Atmospheric, Oceanic and Planetary Physics, University of Oxford, Oxford, UK. [5]Jet Propulsion Laboratory, California Institute of Technology, Pasadena, CA, USA. ✉e-mail: andrew.prata@csiro.au

annotated on the figure panels of Fig. 1, which show the Himawari satellite's view of the eruption at 05:17 UTC. Due to the presence of the main plume over the volcano, at this time (05:17 UTC), the shape of the diffuse cloud could be interpreted as a disk or a ring centred over the volcano. However, if the Himawari data are animated (Supplementary Movie 2), an ice cloud signature in the shape of a ring can be identified (in particular at 05:37 UTC on 15 January), and so we refer to it hereafter as an ice ring. This feature of the eruption has been noted in the literature as a diffuse umbrella cloud likely composed of ice/water at 16–18 km[13,24,25] and has not received a detailed examination. We show here that it is an important (and possibly unique) feature of the Hunga eruption that only seems to have formed under a particularly favourable set of environmental conditions.

## Results and Discussion
### Cloud properties of the ice ring
Quantitative analysis of thermal infrared Himawari-8 data reveals that the ice ring first began to form at around 04:47 UTC and continued to mature for half an hour before reaching its maximum radial extent at around 05:17

UTC (Supplementary Movie 2). By 06:47 UTC the majority of the ice cloud was difficult to discern in satellite data. Figure 1 shows the ice cloud properties at a time when the ice ring had reached its largest radial extent. At this time the ice ring radius (radial distance from the volcano to the outer edge of the ice ring) was ~ 300 km. The shadows in the northeast quadrant of the true colour imagery (depicted in Fig. 1a) indicate that the diffuse cloud was at a lower altitude than the main optically thick umbrella plume centred over the volcano. According to well-established radiative transfer theory, the exceptional signal (at times > 50 K) in the brightness temperature differences (BTDs) between the 8.6 and 11.2 $\mu$m channels (Fig. 1b) indicates that the ice ring was semi-transparent, composed of small ice particles and there was a strong thermal contrast between the surface and the temperature at the height of the cloud[3,26–28]. Typical values for the BTD between channels centred near 8.6 and 11.2 $\mu$m for meteorological ice and liquid-water clouds are ~ 1.5–7 K and ~0.2–2.5 K, respectively, while for clear-sky scenes the BTD is slightly negative due to water vapour absorption[29]. Cloud-top pressure retrievals (see Methods) close to the edges of the ice ring indicate altitudes from ~ 14–17 km (Fig 1c), which broadly agree with geometrically

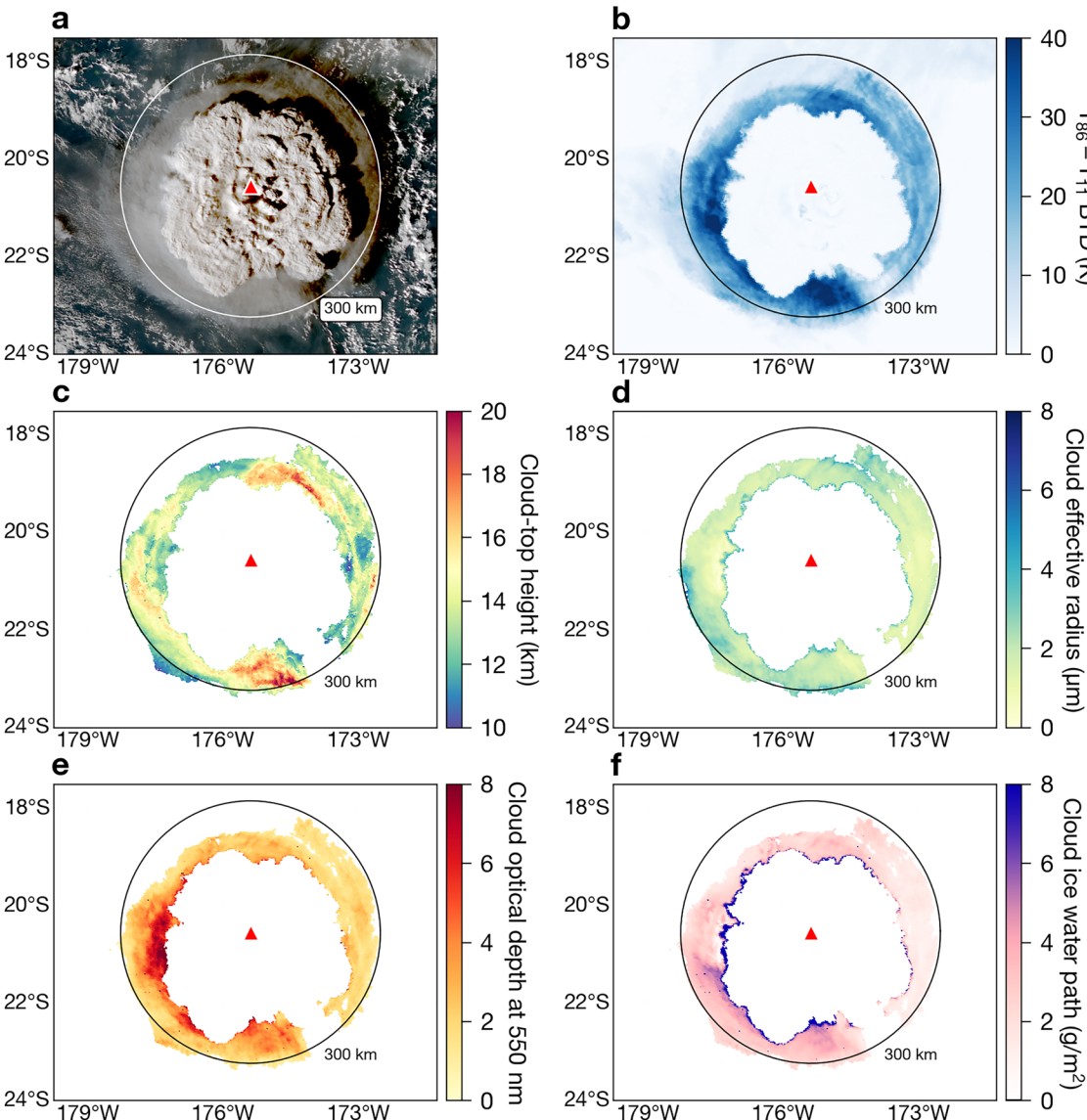

**Fig. 1 | Cloud property retrievals for the ice ring derived from the Advanced Himawari Imager (AHI; aboard the Himawari-8 satellite) at 05:17 UTC on 15 January 2022.** The circle annotated on each panel indicates a radial distance of 300 km from Hunga volcano (red triangle). **a** True colour image. **b** Brightness temperature difference (BTD) between the 8.6 and 11.2 $\mu$m AHI channels. **c** Cloud-top height. **d** Cloud effective radius. **e** Cloud optical depth at 550 nm. **f** Cloud ice water path. For **c**–**f** only pixels with a BTD greater than 15 K are shown. Maps and satellite imagery were generated and processed by the authors with Python 3.9.9 using the Matplotlib 3.5.1 and SatPy 0.33.1 packages.

obtained estimates based on the ice cloud's shadow[cf. 16 km; 24]. The mean ice particle effective radii within the ice ring at 05:17 UTC was ~2 $\mu m$ (Fig 1d). Tropical tropopause cirrus cloud sizes have been reported in the range from 2.5–25 $\mu m$ radius[30]; however, the size observed depends upon the stage of cloud development. The Hunga ice ring particle sizes would fall into the smallest, median ice particle size range (1–6 $\mu m$) reported by Krämer et al.[31,32] who compiled a cirrus ice particle size climatology from in situ aircraft data. Ice particle sizes reported in the Krämer et al.[32] study are based upon the ice crystal mean mass radius, $R_{ice}$, and are comparable to the effective radii satellite retrievals shown in the present study. Krämer et al.[32] show that median $R_{ice}$ values for cirrus clouds at similar altitudes to the ice ring (~15–18 km) lie in the range from 6–18 $\mu m$ and that $R_{ice}$ values for tropical tropopause layer cirrus clouds, at the coldest temperatures (<190 K), are around 8 $\mu m$ (see Fig. 8f of ref. 32).

Ice particle sizes of ~20 $\mu m$ associated with volcanic eruptions have been observed before[3,33] and the particle sizes observed have been attributed to overseeding[i.e. high number concentrations of ice nucleating particles; 33]; however, the ice particles studied here are of an order of magnitude smaller, indicating an earlier stage of ice particle growth that has not previously been documented for ice clouds associated with volcanic eruptions. Such small ice particles are unlikely to be observed under natural conditions. In theory, the smallest ice particles will be observed at the very beginning of the ice particle growth phase during cloud development. Once nucleated, ice particles will initially grow via vapour deposition. Particle growth via vapour deposition is principally driven by the level of supersaturation, and, in the case of heterogeneous ice nucleation, the ice nucleating particle (INP) number concentration and initial (dry) particle size distribution[34–37]. The longer an environment remains supersaturated with respect to ice, the larger an ice particle will be expected to grow. Thus, if the environment is subject to only a short period of supersaturation, then small ice particles may be expected. For distributions of ice particles, low number concentrations will lead to larger particle sizes (and vice versa) as individual ice particles compete for the available background ambient water vapour.

The mean optical depth (at 550 nm) of the ice ring at 05:17 UTC was 2.7 (Fig 1e), which would be classified as opaque cirrus[38] but is considered optically thin at thermal infrared wavelengths (for a 2 $\mu m$ ice particle, an optical depth of 2.7 at 550 nm is equivalent to ~1.8 at 11 $\mu m$). The mean (and standard deviation) ice water path (IWP) was 2.8 ± 1.8 g m$^{-2}$ (Fig 1f) and amounts to a total mass of ice of 0.30 ± 0.09 Tg. This total ice mass is

small in comparison to other ice-rich volcanic plumes[cf. 1–10 Tg; 3,26,39]; however, it is likely that a much greater amount of ice was produced in upper regions of the Hunga plume, which is not the focus of the present study. The ice mass time-series is shown in Supplementary Fig. 1.

## Generation mechanisms

One of the most notable features of the Hunga eruption was the generation of a broad spectrum of atmospheric waves, including the global propagation of surface-trapped Lamb waves[15,40–45] and the first detection of the slower phase speed Pekeris wave[46,47]. Lamb and Pekeris waves represent normal modes of the atmosphere and are preferentially excited by large, impulsive disturbances[46,48–50]. Examples of such impulses include volcanic eruptions[51], nuclear explosions[52], earthquakes[53] and meteor impacts[54]. The global propagation of the atmospheric waves produced by Hunga has been tracked in geostationary satellite data by taking the difference between single-channel brightness temperature measurements at consecutive times. Wright et al.[15] use the 10.4 $\mu m$ wavelength measurements to track the propagation of the Lamb wave, which show temperature changes at the surface. Otsuka[42] used the 6.2 $\mu m$ channel to track the Lamb wave, which responds to mid-tropospheric (~350 hPa or 8 km) temperature changes due to water vapour absorption in this band. Watanabe et al.[47] used the 9.6 $\mu m$ channel to identify the Pekeris wave, which is predicted to exhibit a higher amplitude in the stratosphere and mesosphere than in the troposphere. Our retrievals indicate that the ice ring formed much higher than the mid-troposphere, with cloud-top heights typically reaching 16–18 km, just below and above the local tropopause (~16.5 km). Here we use the 9.6 $\mu m$ channel in combination with the 10.4 $\mu m$ channel to isolate the signal in the lower stratosphere that is due to ozone absorption in this band (see Methods). Figure 2a shows the striking temperature response in the lower–middle stratosphere (LMS) following the 15 January eruption from 04:47 to 04:57 UTC. Figure 2b shows the weighting functions for the 10.4 $\mu m$ and 9.6 $\mu m$ channels. The weighting function is defined as the derivative of the transmittance profile with respect to the vertical coordinate (e.g., height or pressure) for a given wavelength or wavelength interval for an infrared channel on a multi-spectral instrument[55]. Weighting functions therefore, represent the relative contributions of each atmospheric level to the measured top-of-atmosphere radiance in a particular channel. The 9.6 $\mu m$ channel is sensitive to ozone absorption and so its weighting function is tied to the vertical distribution of ozone in the atmosphere[56]. We estimate that the temperature perturbations

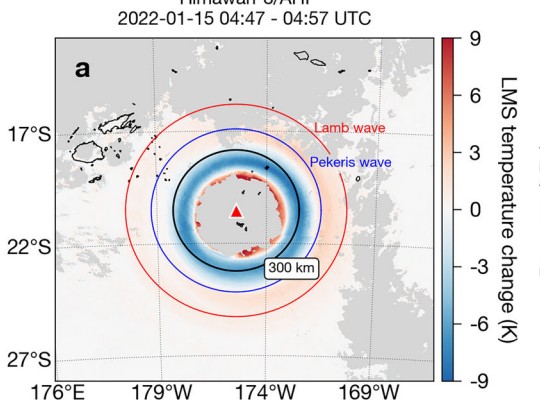

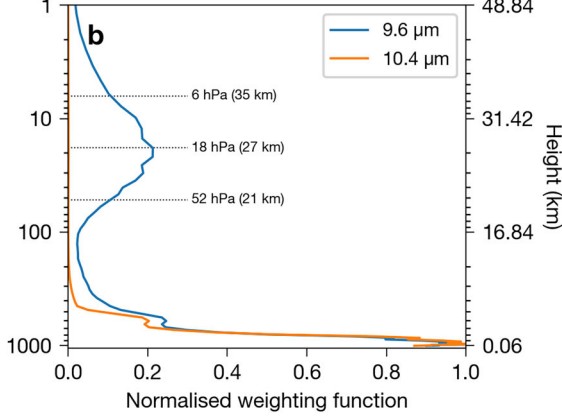

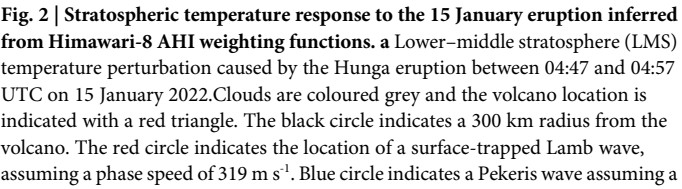

**Fig. 2 | Stratospheric temperature response to the 15 January eruption inferred from Himawari-8 AHI weighting functions. a** Lower–middle stratosphere (LMS) temperature perturbation caused by the Hunga eruption between 04:47 and 04:57 UTC on 15 January 2022. Clouds are coloured grey and the volcano location is indicated with a red triangle. The black circle indicates a 300 km radius from the volcano. The red circle indicates the location of a surface-trapped Lamb wave, assuming a phase speed of 319 m s$^{-1}$. Blue circle indicates a Pekeris wave assuming a phase speed of 245 m s$^{-1}$. Both waves are assumed to be initiated at Hunga volcano at 04:29:30 UTC. **b** Normalised weighting functions for the 9.6 and 10.4 $\mu m$ AHI channels. Weighting functions were computed using RTTOV[Radiative Transfer for TOVS; 85] for a clear-sky, ERA5 profile over the volcano at 04:00 UTC on 15 January 2022. Maps, satellite imagery and plot were generated and processed by the authors with Python 3.9.9 using the Matplotlib 3.5.1 and SatPy 0.33.1 packages.

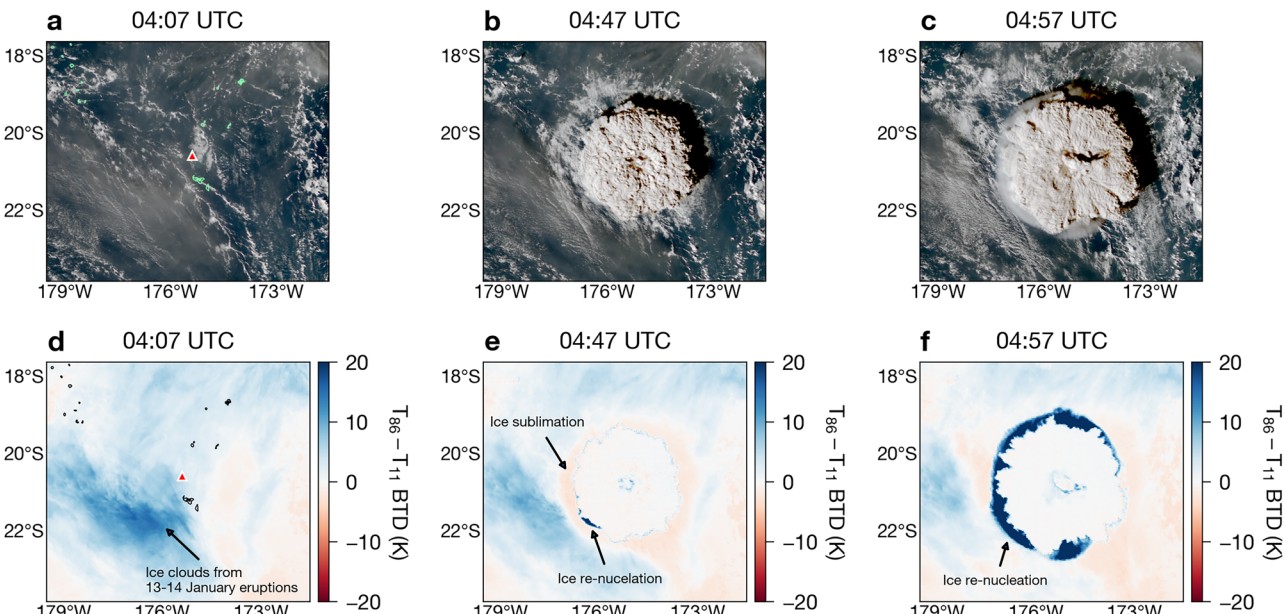

**Fig. 3 | Eruption sequence of the 15 January 2022 Hunga volcano eruption.**
**a–c** AHI true colour imagery (observation times indicated on figure panels).
**d–f** Thermal infrared brightness temperature difference between the 8.6 and 11.2 $\mu$m
AHI channels. Location of Hunga volcano is indicated as a red triangle on **a** and

**d** and coastlines are plotted in green on **a** and black on **d**. Maps and satellite imagery
were generated and processed by the authors with Python 3.9.9 using the Matplotlib
3.5.1 and SatPy 0.33.1 packages.

shown in Fig. 2a would correspond broadly to pressure heights from 6–52
hPa (21–35 km) based on the 9.6 $\mu$m channel weighting function full-width
half-maximum range above the tropopause (Fig. 2b).

The negative temperature anomalies in Figure 2a form a concentric
ring structure. During the first few hours of the eruption these negative
anomalies propagate radially outward and precede the formation of ice
particles (Fig. 1). Figure 2a shows how the temperature response in the
lower-middle stratosphere compares with present estimates of the Lamb
(red circle) and Pekeris (blue circle) wave phase speeds and an assumed
origin time of 04:29:30 UTC which has been derived from the timing and
location of the peak pressure signal in surface pressure stations[15,57]. The
Lamb wave phase speed of 319 m s$^{-1}$ is taken from Jarvis et al.[57] and the
Pekeris wave phase speed of 245 m s$^{-1}$ is taken from Watanabe et al.[47]. The
negative temperature perturbations appear to be consistent with the phase
speed of the Pekeris wave. Here, it is argued that the cold phases of the upper
troposphere-lower stratosphere (UTLS) temperature perturbations caused
by the atmospheric waves associated with the eruption are a plausible
explanation for the generation of the ice ring.

### Environmental conditions for ice formation
In order for ice clouds to form in the atmosphere, there needs to be a source
of liquid water or water vapour that is brought to supersaturation with
respect to ice. For cirrus, recent work suggests that these clouds can be
grouped into two broad categories: in situ origin cirrus and liquid origin
cirrus[31,32,58]. In situ origin cirrus clouds may form due to heterogeneous ice
nucleation or via homogeneous freezing of supercooled solution
aerosols[36,59]. Liquid origin cirrus clouds typically contain larger ice particles
than in situ origin cirrus, as they form from liquid water drops which rapidly
grow once frozen[58,60]. Even in the case of strong updrafts, where liquid water
droplets can survive until they reach ~−38 °C and freeze homogeneously,
the smallest liquid origin cirrus ice particle sizes are typically ~10–20 $\mu$m
radius[31].

Volcanic ash particles are known to be efficient ice nucleating particles
(INPs)[33,61] and it is common in volcanic eruptions for water vapour in the
atmosphere to be entrained into the plume, condense to liquid water and
freeze in updrafts during transport to plume-top[62,63]. Magmatic water,
expelled during an eruption, is also a source of water for ice nucleation,

meaning that volcanogenic ice can form in relatively dry atmospheres[39]. A
further source of water vapour can be added to the atmosphere via phrea-
tomagmatic interactions[3]. It is likely that all of the aforementioned
mechanisms were at play during the Hunga eruption, raising the question:
Which ice nucleation pathway was responsible for the formation of the
ice ring?

Given the size of the ice particles (~2 $\mu$m), potential availability of
efficient (volcanic) INPs and the similarity between the temperature
anomaly at 04:57 UTC (Fig. 2a) and the ice ring at 05:17 UTC (Fig. 1),
we explored the hypothesis that the temperature perturbation gener-
ated by atmospheric waves in this altitude region triggered rapid, large-
scale heterogeneous deposition nucleation in the UTLS. We also show
that the source of ambient water vapour and INPs in the vicinity of
Hunga volcano on 15 January came from the 13–14 January eruptive
activity.

Figure 3 shows a sequence of true colour images and the corresponding
BTDs between the 8.6 and 11.2 $\mu$m channels during the initial stages of the
15 January eruption. At 04:07 UTC, before the umbrella plume had estab-
lished itself, semi-transparent ice clouds (BTDs > 0 K) are present over
Hunga volcano and cover most of the surrounding area except for some
parts of the southeast (identified by negative BTDs; Fig. 3a, d). Analysis of
AHI data prior to 04:07 UTC shows that these ice clouds were produced by
eruptive activity during 13–14 January (see Supplementary Movie 1). By
04:47 UTC, a ring of negative BTDs had surrounded the main umbrella
(Fig. 3b, e) and a small but notable region showed extreme positive BTDs.
Negative BTDs indicate the presence of water vapour[29], which provides
compelling evidence that the pre-existing ice clouds had been sublimated (in
the form of a ring) by 04:47 UTC. Ten minutes later, however, extreme
positive BTD values (maximum BTD of 48.5 K at this time) had almost
completely replaced the negative BTDs from the previous observation time,
indicating that numerous INPs were being re-nucleated by 04:57 UTC
(Fig. 3c, f). At 05:17 UTC the ice ring reached its largest radial extent (Fig. 1)
and subsequently dissipated thereafter.

### Evolution of temperature in the ice-ring region
To further investigate the ambient conditions for ice formation, we selected
a spatial region between two concentric circles corresponding to radial

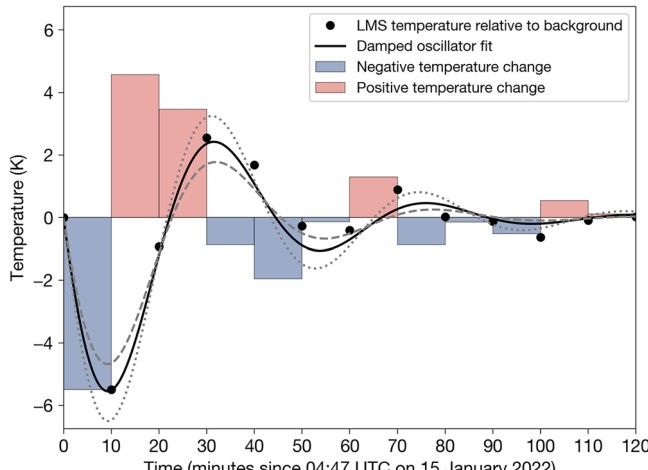

**Fig. 4 | Satellite-based estimation of the temporal evolution of temperature in the lower-middle stratosphere (LMS; 6–52 hPa; 21–35 km) within the ice-ring region (defined as the region enclosed between two concentric circles with radial distances of 250 and 300 km from Hunga volcano).** Red and blue bars indicate differences in temperature between satellite observation time steps (i.e., temperature perturbations). Black data points represent the LMS temperature change with time relative to background temperature (computed as the cumulative sum of the temperature perturbations; see Methods for details). The black line indicates a damped harmonic oscillator model fit and the grey dotted and dashed lines indicate the model fit uncertainty (see Methods for details). The plot was generated by the authors with Python 3.9.9 using the Matplotlib 3.5.1 package.

distances of 250 and 300 km from the volcano (i.e., the annulus where the majority of the ice ring formed), referred to hereafter as the ice-ring region. The analysis, based on the AHI weighting functions (Fig. 2), shows that there was a temperature reduction in the ice-ring region at altitudes from 6–52 hPa (21–35 km). The introduction of an abrupt pressure pulse at the surface is theorised to cause a change in temperature throughout the entire atmospheric profile and the magnitude of this temperature change increases exponentially with height[42,48,50,54]. Therefore, it is expected that the magnitude of the temperature change due to the Hunga eruption would be lower at altitudes where the ice ring formed (75–115 hPa; 16–18 km).

Although the precise amplitude of the temperature perturbation at these altitudes is difficult to determine, Fig. 4 shows both the temperature perturbation time series (red and blue bars) and the temperature change with time relative to the background temperature (black data points). The term background temperature is used to refer to the temperature of the LMS in the ice-ring region at 04:47 UTC (i.e., before the large cold-ring anomaly, shown in Fig. 2a, is detected). What is immediately apparent is that the temperature time series can be well described by the equation for a damped harmonic oscillator (black solid line in Fig. 4).

### Ice particle modelling

To investigate the effect of temperature oscillations on ice cloud formation, we conducted a series of ice particle growth simulations. The ice particle model developed in the present study simulates growth via vapour deposition for a size distribution of spherical particles in a unit volume (see Methods). To constrain the problem, it is assumed that the initial INP size distribution was lognormal with a mean of 0.1 $\mu m$ and spread of 1.5 and the number concentration was 0.1 cm$^{-3}$. The size distribution and its properties were informed by observations of volcanic ash particles[64,65] and studies showing that ash particles can act as efficient INPs[33,61]. To capture uncertainty in the modelled ice particle size, other model parameters were allowed to vary. Specifically, an ensemble of model simulations were conducted at four initial levels of ice supersaturation (0.1, 1, 5 and 10 %), with varying initial ambient conditions (temperature from ~ −82 to −78 °C, pressure from ~75 to 115 hPa and height from ~16 to 18 km corresponding to five

ERA5 model levels), UTLS temperature perturbations (minimum initial amplitudes from −2 to −0.4 K in steps of 0.4 K) and ice mass accommodation coefficients (from 0.05 to 0.30 in steps of 0.05;[66]), resulting in 150 simulations at each initial supersaturation level. In each model simulation, the ice supersaturation was allowed to evolve with time according to Eqs. (16)–(23) (see Methods). Finally, for comparison, the same set of model experiments was performed, but for the case where the ambient temperature was held constant in time (rather than evolving as a damped harmonic oscillator).

Figure 5 shows how the mean effective radius of the modelled ice particles compares with the satellite-retrieved mean effective radius within the ice-ring region. Note that the error bars on Fig. 5 represent the standard deviation of the satellite-retrieved effective radii and therefore are representative of the variation in effective radii within the ice-ring region. Supplementary Fig. 2 provides the corresponding mean effective radius uncertainty determined from the satellite optimal estimation algorithm (see Methods). The relative uncertainties for the satellite-retrieved effective radii range from 14–36%. Additionally, as the optical depth increases, the uncertainty on the effective radius decreases and vice-versa, which is expected for thermal infrared optimal estimation retrieval algorithms applied to semi-transparent ice clouds[67,68].

The ice particle modelling results demonstrate that a model for depositional growth of a distribution of ice particles can reproduce the particle sizes inferred from the satellite measurements but only under certain model assumptions. For initial supersaturations of 0.1 and 1 %, the constant ambient temperature simulations (red dashed lines; Figs. 5a and b) systematically underestimate particle size compared to the satellite retrievals. The corresponding damped harmonic oscillator simulations (black solid lines; Figs. 5a and b) do capture some of the variation in particle size at the beginning of the simulation, but after 30 minutes, they too underestimate particle sizes for the remainder of the simulation. For initial supersaturations of 5 and 10 %, a better agreement is found for the constant ambient temperature simulations (red dashed lines; Figs. 5c and d) compared to the lower initial supersaturation runs (red dashed lines; Figs. 5a and b); however, in the case of the 5 % initial supersaturation runs, particle sizes are generally underestimated until the end of the simulation and in the case of the 10 % initial supersaturation runs particle sizes are overestimated after ~ 50 minutes. In general, the model simulations that include the temperature wave perturbation (damped harmonic oscillator model) better capture the initial ice particle sizes (~20 minutes into the simulation) compared to the counterpart runs that assume a constant ambient temperature. The best agreement between model simulations and satellite retrievals is found when the temperature wave perturbation is included and the initial supersaturation is set to 5 % (black solid line; Fig. 5c). In this case, the ensemble mean of the effective radii model simulations remains within the standard deviations of the satellite-retrieved effective radii within the ice-ring region during the 2-h time period from 04:47–06:47 UTC. In addition, including the temperature wave perturbation reproduces what appears to be oscillations in the satellite-retrieved effective radius with time; however, on the basis of the present analysis, this inference is speculative, as the standard deviations of the satellite-retrieved effective radii are larger than the variations of mean effective radii with time.

### Significance of the ice ring

To our knowledge, the ice ring has not been documented for a volcanic eruption before. Due to its transient nature, this is perhaps not surprising and highlights the importance of geostationary satellite observations with 10 minute (or better) temporal resolution. An important conclusion here is that the ice ring should not be confused with an umbrella plume when trying to infer eruption rates. We propose that the ice ring formed as a result of an atmospheric wave (possibly a Pekeris wave) generated by the paroxysmal eruption of Hunga volcano on 15 January 2022. The atmospheric wave triggered large-scale subsidence in the remanent ice clouds left behind by eruptive activity during 13–14 January, leaving the atmosphere rich with

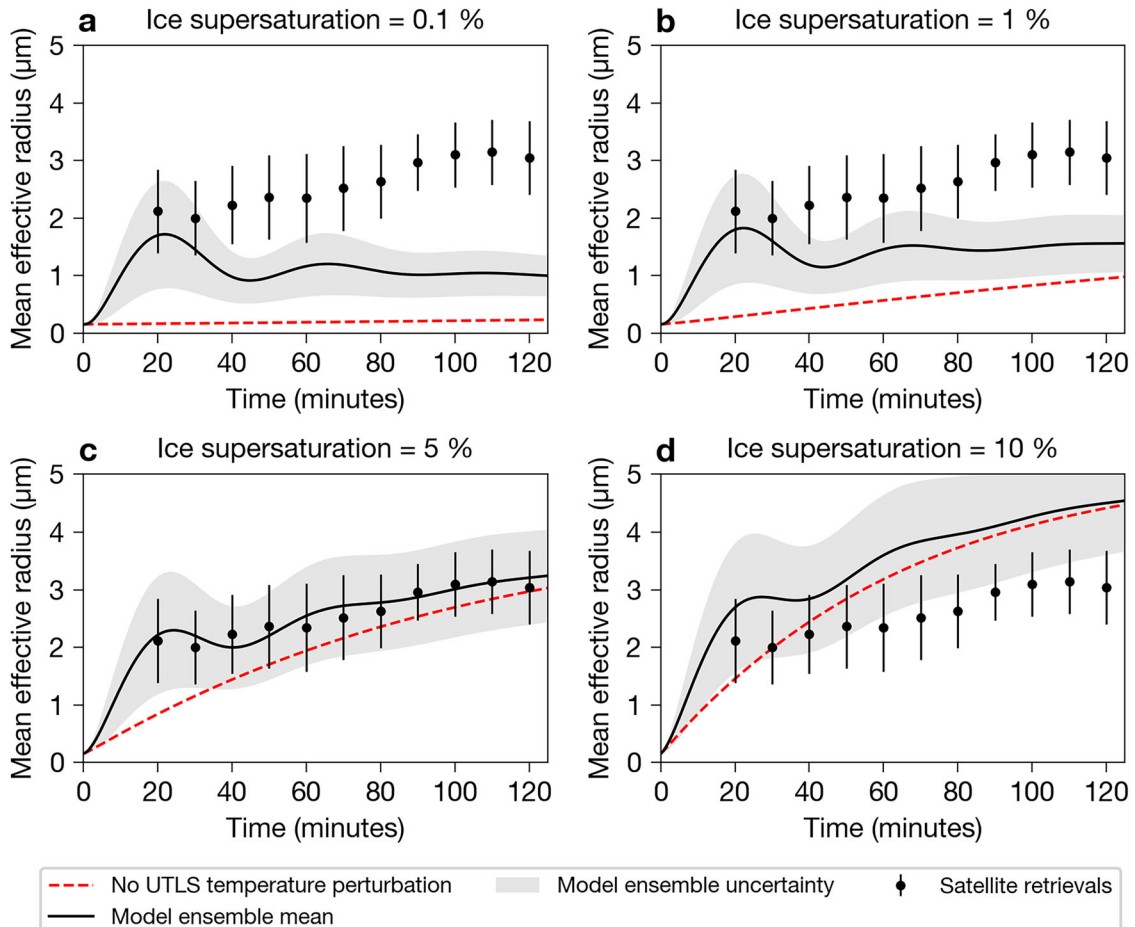

**Fig. 5 | Comparison of satellite retrievals with ensemble modelling of mean ice particle effective radius in the ice ring.** Evolution of the mean ice particle effective radius over time for the ice ring for supersaturation of **a** 0.1 %, **b** 1 %, **c** 5 % and **d** 10 %. Time is calculated in minutes relative to 04:47 UTC on 15 January 2022. Black data points indicate the spatial mean and standard deviation (error bars) of satellite effective radius retrievals within the ice-ring region (region enclosed between two concentric circles of radial distances from 250 to 300 km). The black solid lines indicate the ensemble mean and standard deviation (shaded regions) of the effective radius from the model simulations that include a damped harmonic oscillator temperature perturbation. The red dashed lines represent the mean effective radius of the ensemble for runs with a constant ambient temperature (further details of the model simulations can be found in the Methods section). The plot was generated by the authors with Python 3.9.9 using the Matplotlib 3.5.1 package.

ice-nucleating particles and in a supersaturated state. The UTLS temperature then evolved as a damped harmonic oscillator, resulting in widespread ice re-nucleation in the form of a ring, permitting satellite observations of ice nucleation at the early stages of cloud development. The record positive brightness temperature differences reported here can be explained by the geostationary satellite measurements capturing the initial stages of ice particle growth via vapour deposition where the smallest ice particle sizes are expected. Based on satellite observations, we have found that the maximum extent of the ice ring was 300 km from source and formed at 75–115 hPa (16–18 km; just below and above the local tropopause ~16.5 km). The satellite observations, together with our modelling results, indicate that the presence of pre-existing ice clouds (and INPs) were necessary to explain the supersaturated conditions required to form the ice ring. As shown by Fig. 5, a temperature perturbation in the UTLS modelled as a damped harmonic oscillator with an initial ice supersaturation of ~5% can reproduce the small (~2 μm) mean effective radii retrievals found in the annulus region bounded between radial distances from 250–300 km from the volcano. The fact that pre-existing ice clouds did not completely cover the region over Hunga volcano may, in part, explain why the ice ring did not form in a perfectly circular shape around the volcano. Given the specific set of environmental conditions required for the ice ring to form, the Hunga eruption case study serves as a natural experiment for which more sophisticated ice nucleation models can be tested.

## Methods
### Cloud property retrievals
To retrieve ice cloud properties from satellite measurements we use the Optimal Retrieval of Aerosol and Cloud (ORAC) algorithm[69–71]. ORAC is a cloud and aerosol retrieval code that uses optimal estimation to estimate cloud properties (more formally, state variables) from satellite measurements. The satellite measurements used here are taken from the Japanese Meteorological Agency's geostationary instrument, the Advanced Himawari Imager (AHI), aboard the Himawari-8 satellite[72]. The state variables retrieved in the present study include the surface temperature, ice cloud-top pressure, effective radius and optical depth (at 550 nm). Ice water path is computed from the ice cloud effective radius and optical depth[see 69, for details]. Recent work on volcanic ash clouds has demonstrated the use of both the 10.4 and 13.3 μm channels for improving retrievals of effective radius and cloud-top height, respectively[71]. For ice clouds, several authors have shown the importance of the 8.6, 11 and 12 μm channel combination for the simultaneous retrieval of effective radius and optical depth[67,68,73,74]. Wang et al.[67] have shown that the uncertainty for thermal infrared retrieval algorithms reduces as the number of measurements increase and show that the uncertainty in the retrieved effective radius will be large for optically thick ($\tau \sim 10$) and optically thin ($\tau \sim 0.1$) ice clouds with the lowest uncertainties expected for semi-transparent ice clouds ($\tau \sim 1$–5). In the present study, we use the 8.6, 10.4, 11.2, 12.4 and 13.3 μm channels to retrieve cloud properties

from thermal infrared measurements. This retrieval setup has the advantage of retrieving microphysical information during the day and night, which is relevant for the Hunga eruption as it occurred as the Sun was setting, necessitating thermal infrared retrievals. In the ORAC retrieval code, ice particles are assumed to be spheres, conform to a gamma size distribution and have a complex refractive index given by Warren and Brandt[75]. The gamma distribution can be written in terms of the effective radius ($r_e$) and effective variance ($v_e$), and normalised so that $a$ represents the number of particles per unit volume:

$$n(r) = \frac{a}{(r_e v_e)^{\frac{1-2v_e}{v_e}} (\frac{1-3v_e}{v_e})!} r^{\frac{1-3v_e}{v_e}} \exp\left(-\frac{r}{r_e v_e}\right), \qquad (1)$$

where $n(r)$ represents the size distribution of particles with radius $r$. In the present study, $a = 1$ and $v_e = 0.111$ and $r_e$ varies over a range of values to construct a look-up table. The value for $v_e$ and choice of the size distribution used in the present study follows that of the Moderate-Resolution Imaging Spectroradiometer (MODIS) cloud retrieval algorithm collection 6, which has been informed by radiative transfer model sensitivity studies and numerous field campaigns[see 76,for details]. Further details of the ice microphysical model, ancillary data processing and forward model equations used within ORAC can be found in McGarragh et al.[70].

The ORAC retrieval is run on every pixel in a given scene and therefore assumes that each pixel contains ice (with a cloud fraction of 1). To ensure this assumption is valid when considering the mean effective radii within the ice-ring region (black data points with error bars in Fig. 5), we applied four criteria to mask out pixels unrelated to the ice ring itself:

1. We mask all pixels with an 8.6 - 11.2 $\mu$m BTD less than 0, which aims to remove clear-sky pixels, water vapour pixels and ash-contaminated pixels.
2. We mask all pixels with an effective radius greater than 4 $\mu$m, which aims to remove larger ice particles in the scene that are not related to the ice ring.
3. We mask all pixels with a 10.4 $\mu$m brightness temperature of less than 270 K, which serves to remove the main umbrella cloud and lower-level optically thick cloud in the scene (see greyed-out regions in Fig. 2a).
4. After applying criteria (1)–(3), if there are fewer than 50 remaining valid pixels, then the mean effective radius is discarded (i.e., not considered a valid representation of the mean ice cloud properties of the ice ring).

The main effect of criterion (4) is to remove the mean effective radii at 04:47 UTC and 04:57 UTC. At these times, the ice particle radius retrievals represent ice particles from the pre-existing clouds produced by the eruptive activity on 13–14 January and are therefore not related to the ice ring being modelled in the present study. At the largest radial extent of the ice ring, the mean effective radius is calculated from ~7000 valid pixels.

## Visualising atmospheric waves with a digital filter

Wright et al.[15] have shown that a single-channel (10.4 $\mu$m) brightness temperature difference between two consecutive time steps can be used to track the propagation of the Lamb wave. However, the temperature change found using this method cannot be directly used to infer temperature changes near 100 hPa (16.5 km), where the ice ring formed, because the 10.4 $\mu$m weighting function peaks near the surface. Otsuka[42] studied the use of the 6.2 $\mu$m channel from AHI to track the Lamb wave. Otsuka showed that the second derivative of the difference in time was an effective method for visualising atmospheric wave propagation because the second derivative highlights signals propagating at faster speeds than other slowly changing signals in the atmosphere. The 6.2 $\mu$m weighting function peaks near 8 km and, again, does not represent the region in the atmosphere where our retrievals indicate that the ice ring formed. While it would be ideal to have a weighting function peak in the UTLS (where most of the ice cloud appears to have formed), all AHI weighting functions have low sensitivity near this

level in the atmosphere. Most thermal AHI channels respond to either the surface (window channels) or to water vapour (which reaches a minimum near the tropopause). The exceptions are the 13.3 and the 9.6 $\mu$m channels whose weighting functions are mainly influenced by $CO_2$ and $O_3$ absorption, respectively. Watanabe et al.[47] used the 9.6 $\mu$m (ozone) channel to identify the Pekeris wave, which is a slower-moving atmospheric wave 180° out of phase with the surface-trapped Lamb wave. However, as the 9.6 $\mu$m weighting function is also sensitive to the surface, the temperature signal determined from the difference in time of the 9.6 $\mu$m brightness temperature will include both signals originating at the surface and the lower–middle stratosphere (LMS) where ozone concentrations peak.

To alleviate this problem and improve upon the method presented by Watanabe et al.[47], we took finite differences in time between the 9.6 $\mu$m and 10.4 $\mu$m brightness temperatures and then subtracted the 10.4 $\mu$m finite difference from the 9.6 $\mu$m finite difference. The end result can be thought of as a kind of digital filter, optimised to be sensitive to the LMS. The digital filter, $T'_{LMS}(t_i)$, is constructed as follows

$$T'_{LMS}(t_i) = T'_{9.6}(t_i) - f_W T'_{10.4}(t_i), \qquad (2)$$

where $t$ indicates observation time, $i$ indicates the observation time step and $f_W$ is a weighting factor. $T'_{9.6}(t_i)$ and $T'_{10.4}(t_i)$ represent backward finite differences in time and are defined as

$$T'_{9.6}(t_i) = \frac{T_{9.6}(t_i) - T_{9.6}(t_{i-1})}{t_i - t_{i-1}}, \qquad (3)$$

and

$$T'_{10.4}(t_i) = \frac{T_{10.4}(t_i) - T_{10.4}(t_{i-1})}{t_i - t_{i-1}}. \qquad (4)$$

The weighting factor, $f_W$, is determined by varying $f_W$ from 0 to 1 (in steps of 0.01) until the variance of $T'_{LMS}$ for a regional area centred over Hunga volcano is minimised. The weighting factor was introduced to account for the variation in the brightness temperature difference (between the 9.6 and 10.4 channels) with satellite zenith angle (due to limb darkening) and to reduce the signals of cloud motion between time steps. For the region over Hunga volcano, we found $f_W = 0.57$. Once $T'_{LMS}$ is determined for each AHI pixel, we mask out pixels containing opaque clouds that can interfere with the extracted temperature signal. Specifically, we mask pixels where $T_{9.6}$ is less than 260 K and where $T_{10.4}$ is less than 270 K (an example of the opaque cloud masking is shown in Fig. 2a).

## Damped harmonic oscillator temperature model

To generate the time-series shown in Fig. 4, we took an Eulerian point-of-view and considered an area enclosed between two circles centred on the volcano at radial distances of 250 km and 300 km (referred to as the ice-ring region). All pixels outside of the ice-ring region were masked and the mean of the remaining unmasked pixels was calculated to produce $\overline{T'_{LMS}}$ at each observation time step (i.e., red and blue bars in Fig. 4). However, since $\overline{T'_{LMS}}(t)$ represents the rate of change in temperature with time, we approximated the integral of $\overline{T'_{LMS}}(t)$ with Riemann sums and cumulative integration to derive the temperature time-series (i.e., black data points in Fig. 4) as

$$\overline{T_{LMS}}(t_i) = \sum_{j=1}^{i} \overline{T'_{LMS}}(t_j)(t_j - t_{j-1}) \, for \, i = 1, 2, \ldots, N, \qquad (5)$$

where $N$ is the total number of time-steps (AHI observations) considered for the analysis. Note that as the recovered temperature time-series is derived based on finite differences, it represents temperature change relative to the LMS temperature at the first observation time (i.e. at $i = 0$, $\overline{T_{LMS}}(t_0) = 0$ K). In the present study, $t_0$ is set to 04:47 UTC on 15 January 2022.

**Table 1 | Values of constants used in the ice particle growth model not reported in the text**

| Constant | Symbol | Value | Units | Reference |
|---|---|---|---|---|
| Latent heat of sublimation[1] | $L_s$ | $2.836 \times 10^6$ | J kg$^{-1}$ | Feistel and Wagner[82] |
| Gas constant of water vapour | $R_v$ | 461 | J kg$^{-1}$K$^{-1}$ | Romps[83] |
| Gas constant of dry air | $R_d$ | 287.04 | J kg$^{-1}$K$^{-1}$ | Romps[83] |
| Specific heat of dry air at constant volume | $c_v$ | 718 | J kg$^{-1}$K$^{-1}$ | Romps[83] |
| Specific heat capacity of air at constant pressure | $c_p$ | 1004 | J kg$^{-1}$K$^{-1}$ | Wallace and Hobbs[84] |
| Thermal accommodation coefficient | $\alpha_T$ | 1 | dimensionless | Skrotzki et al.[77] |

[1] Value used is valid for 190K < $T$ < 273K.

Since we do not know the precise amplitude of the temperature change in the UTLS, we scale $\overline{T_{LMS}(t_i)}$ by a factor $f_T$, so that the minimum amplitude is consistent with a range of values from −2 to −0.4 K. We then add the ambient temperature, $T(p)$, at a given ERA5 pressure level, $p$, to estimate the temperature time-series at $p$ as follows

$$\overline{T_p(t_i)} = f_T \overline{T_{LMS}(t_i)} + \overline{T_p(t_0)}, \tag{6}$$

where $f_T$ is a scaling factor and $\overline{T_p(t_0)}$ denotes the mean ambient temperature at some pressure level, $p$, within the ice ring region before the passage of atmospheric wave. In our ice particle modelling experiments, we consider five different pressure levels spanning from 115–75 hPa (16–18 km). Finally, so that the temperature can vary at any temporal resolution, we fit $\overline{T_p(t)}$ (which is determined at 10-minute intervals), to the equation for a damped harmonic oscillator. Specifically, we fit the following equation to $\overline{T_p(t)}$ using the Levenberg-Marquardt (LM) algorithm:

$$T_{fit}(t_i) = Ae^{-\zeta t_i} \sin(\omega t_i), \tag{7}$$

where $Ae^{-\zeta t_i}$ represents the amplitude of the temperature wave, which dampens with time $t$ according to the damping coefficient $\zeta$ and $\omega$ is angular frequency. Uncertainties in the LM minimisation were computed by taking the square root of the diagonal of the error covariance matrix.

**Ice particle growth model**

To test the hypothesis that an adiabatic change in temperature induced by the Hunga eruption was sufficient to produce small ice particles in the UTLS, we devised a model that grows a distribution of ice particles via vapour deposition in a unit volume of air. We approach the problem of the growth of a population of ice particles by numerically solving the growth equation for each ice particle in the distribution and the equation for the rate change of supersatuation.

The growth equation for a single, spherical ice particle of radius $r$, including kinetic effects, is given by[34,35,37]

$$r\frac{dr}{dt} = \frac{S_i - 1}{\left(\frac{L_s}{R_v T} - 1\right)\frac{L_s \rho_i(T)}{KTf(\alpha_T)} + \frac{\rho_i(T)R_v T}{De_{s,i}(T)g(\alpha_m)}}, \tag{8}$$

where $S_i = e/e_{s,i}$ is the saturation ratio with respect to ice, $L_s$ is the latent heat of sublimation, $R_v$ is the gas constant of water vapour, $T$ is the ambient temperature, $K$ is the thermal conductivity of air, given by[37]

$$K = 4.1868 \times 10^{-3}[5.69 + 0.017(T - T_0)], \tag{9}$$

where $T_0 = 273.15$ K, $D$ is the water vapour diffusion coefficient in air given by[37]

$$D = 2.11 \times 10^{-5}\left(\frac{T}{T_0}\right)^{1.94}\left(\frac{p_0}{p}\right), \tag{10}$$

where $p_0 = 1013.25$ hPa. The density of ice as a function of temperature ($\rho_i(T)$; g cm$^{-3}$) is computed using the quadratic fit from Table 1 of Skrotzki et al[77]:

$$\rho_i(T) = a + bT + cT^2, \tag{11}$$

where $a = 0.9167$ g cm$^{-3}$, $b = -1.75 \times 10^{-4}$ g cm$^{-3}$°C$^{-1}$, $c = -5 \times 10^{-7}$ g cm$^{-3}$ °C$^{-2}$ and $T$ is temperature in °C.

The terms $f(\alpha_T)$ and $g(\alpha_m)$ in Eq. (8) are the kinetic correction terms[78]. The $f(\alpha_T)$ normalisation factor is given by

$$f(\alpha_T) = \frac{r}{r + l_{\alpha_T}}, \tag{12}$$

with

$$l_{\alpha_T} = \left(\frac{K}{\alpha_T p}\right)\frac{(2\pi R_d T)^{1/2}}{c_v + R_d/2}, \tag{13}$$

where $R_d$ is the gas constant of dry air, $c_v$ is the specific heat of dry air at constant volume and $\alpha_T$ is the thermal accommodation coefficient.

The $g(\alpha_m)$ normalisation factor is given by

$$g(\alpha_m) = \frac{r}{r + l_{\alpha_m}}, \tag{14}$$

with

$$l_{\alpha_m} = \frac{D}{\alpha_m}\left(\frac{2\pi}{R_v T}\right)^{1/2} \tag{15}$$

where $\alpha_m$ is the ice mass accommodation coefficient.

The change in supersaturation with time can be determined as[35]

$$\frac{dS_i}{dt} = P - C, \tag{16}$$

where $P$ is a production term and $C$ a condensation term (or in our case a deposition term). The $P$ and $C$ terms are commonly defined by considering a rising parcel of air (with vertical velocity $dz/dt$), where the saturation ratio increases as the parcel rises but reduces as more liquid water drops (or ice particles) form and contribute to increases in the liquid water (or ice) mass mixing ratio:

$$\frac{dS_i}{dt} = Q_1\frac{dz}{dt} - Q_2\frac{dw_i}{dt}, \tag{17}$$

where $dw_i/dt$ describes the rate change in the ice mass mixing ratio ($w_i$). The $Q$ terms are defined as follows

$$Q_1 = \frac{1}{T}\left(\frac{\epsilon L_s g}{R_d c_p T} - \frac{g}{R_d}\right), \tag{18}$$

where $c_p$ is the specific heat capacity of air at constant pressure, $g = 9.81$ m s$^{-2}$ is the acceleration due to gravity, $\epsilon = M_w/M_d = 0.622$, $M_d = 28.9644$ g mol$^{-1}$ is the molar mass of dry air and $M_w = 18.0153$ g mol$^{-1}$ is the molar mass of $H_2O$ and

$$Q_2 = \rho_a \left( \frac{R_d T}{\epsilon e_{s,i}} + \frac{\epsilon L_s^2}{pTc_p} \right), \quad (19)$$

where $\rho_a$ is the density of air and is given by

$$\rho_a = \frac{p}{R_d T_v}, \quad (20)$$

where $T_v = T(1 + w/\epsilon)/(1 + w)$ is the virtual temperature and $w$ is the water vapour mass mixing ratio.

In our case, we assume that the updraft velocities in the ice-ring region are negligible ($dz/dt \approx 0$) and instead evolve the ambient temperature according to the fit found for the damped harmonic oscillator equation given in Eq. (7). This approach leads us to re-write Eq. (17) as follows:

$$\frac{dS_i}{dt} = \frac{dS_{fit}}{dt} - Q_2 \frac{dw_i}{dt}, \quad (21)$$

where $S_{fit}(t)$ describes the change in supersaturation due to the change in temperature according to the fitted equation for the damped harmonic oscillator:

$$S_{fit}(t) = \frac{e}{e_{s,i}(T_{fit}(t))}. \quad (22)$$

The rate change of the ice mass mixing ratio will change according to the growth of ice particles in the distribution. Specifically, it can be determined as[34]

$$\frac{dw_i}{dt} = \frac{4}{3}\pi \frac{\rho_i}{\rho_a} \frac{d}{dt} \sum_{j=0}^{N} r_j^3, \quad (23)$$

where $N$ is the total number of ice particles in the distribution per unit volume of air and $r_j$ represents the radius of the $j$th particle in the distribution. To simulate this, we randomly sample $1 \times 10^5$ particles (equivalent to a number concentration of 0.1 cm$^{-3}$) from a lognormal distribution with a mean of 0.1 $\mu$m and spread of 1.5[64,65]. For each particle, $r_j$, we numerically solve Eq. (8), which allows us to evaluate the summation in Eq. (23) and update $w_i$ at each time step.

Physically, our model allows supersaturation to increase or decrease according to changes in ambient temperature governed by the damped harmonic oscillator equation and accounts for the decrease in supersaturation due to the production of a distribution of ice particles. Note that in the case of the constant ambient temperature simulations $dS_{fit}/dt = 0$, and so the ambient supersaturation ($S_i$) can only decrease from its initial value.

## Data availability
The Himawari-8 level 1b HSD (Himawari Standard Data) dataset used for the present analysis (Figs. 1, 2 and 3) was accessed from the JAXA's P-Tree system and is freely available via their ftp site (https://www.eorc. jaxa.jp/ptree/userguide.html, last access: 8 September 2023). The ERA5 reanalysis products[79] were accessed from the Centre for Environmental Data Analysis archive on JASMIN [https://catalogue.ceda.ac.uk/uuid/ f809e61a61ee4eb9a64d4957c3e5bfac, last access: 8 September 2023;][80]. The ORAC ice particle retrieval files (Fig. 1), ice ring times-series data (Figs. 4 and 5) and ice particle modelling ensemble output (Fig. 5) can be accessed from the Oxford University Research Archive (ORA; https:// ora.ox.ac.uk).

## Code availability
The code used for the cloud property retrievals is the Optimal Retrieval of Aerosol and Cloud (ORAC). The code is open source and can be accessed from GitHub [https://github.com/ORAC-CC/orac, last access: 8 September 2023;][69,70,81]. The code developed for the ice particle modelling can be made available upon request by contacting Dr Andrew T. Prata (andrew.prata@csiro.au).

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

## Acknowledgements

A.T.P. and R.G.G. acknowledge funding from the Natural Environment Research Council (NERC) R4Ash project (NE/S003843/1). I.A.T. and R.G.G. were supported by the NERC Centre for Observation and Modelling of Earthquakes, Volcanoes, and Tectonics (COMET), a Leverhulme Trust Research Project Grant and the NERC project VPLUS (NE/S004025/1). This study was partly funded through NERC's support of the National Centre for Earth Observation. The research carried out at the Jet Propulsion Laboratory, California Institute of Technology, was performed under a contract with the National Aeronautics and Space Administration. We thank Dr Fred Prata for many stimulating discussions on the 'ice ring', which helped shape our early hypothesis. We thank three anonymous reviewers for their thorough and detailed reviews, which led to significant improvements to the manuscript.

## Author contributions

A.T.P. conceived the study, led the writing of the manuscript, conducted the satellite ice particle retrievals and analysis and developed the ice particle model. R.G.G. developed the Mie scattering routines for the ice particle satellite retrievals, analysed results, provided scientific interpretation and contributed to writing the manuscript. I.A.T. analysed results, provided scientific interpretation and contributed to writing the manuscript. A.L. contributed to developing the ice particle model, scientific analysis and contributed to writing the manuscript.

## Competing interests

The authors declare no competing interests.
