## [Transparent Peer Review file · Communications Earth & Environment]

Transient ice ring observed during the 15 January 2022 eruption of Hunga volcano

Corresponding Author: Dr Andrew Prata

Version 0:

Decision Letter:

Dear Dr Prata,

I hope you are well?

Your manuscript titled "Transient ice ring observed during the 15 January 2022 eruption of Hunga volcano" has now been seen by 3 reviewers, and we include their comments at the end of this message. They find your work very much of interest, but some important points are raised. We are interested in the possibility of publishing your study in Communications Earth & Environment, but would like to consider your responses to these concerns and assess a revised manuscript before we make a final decision on publication.

We therefore invite you to revise and resubmit your manuscript, along with a point-by-point response that takes into account the points raised. Please highlight all changes in the manuscript text file. As you revise your manuscript, please also consider the following editorial thresholds in addition to the reviewers' comments:

- Provide compelling evidence that the simplified modelling approach to ice formation produces realistic results that can explain observations.
- Ensure that all assumptions are explained fully and that propagated uncertainties are acknowledged, and expressed explicitly where possible.
- Present comprehensive documentation of the initial observations to avoid ambiguity when comparing modelled to observed results.

Please use the following link to submit your revised manuscript, point-by-point response to the referees' comments (which should be in a separate document to any cover letter), a tracked-changes version of the manuscript (as a PDF file) and the completed checklist:

Link Redacted

We hope to receive your revised paper within six weeks; please let us know if you aren't able to submit it within this time so that we can discuss how best to proceed. If we don't hear from you, and the revision process takes significantly longer, we may close your file. In this event, we will still be happy to reconsider your paper at a later date, as long as nothing similar has been accepted for publication at Communications Earth & Environment or published elsewhere in the meantime.

Please do not hesitate to contact us if you have any questions or would like to discuss these revisions further. We look forward to seeing the revised manuscript and thank you for the opportunity to review your work.

Best regards,

Emma Nicholson, PhD
Editorial Board Member
Communications Earth & Environment
orcid.org/0000-0003-1749-9285

Joe Aslin
Senior Editor
Communications Earth & Environment

EDITORIAL POLICIES AND FORMATTING

Editorial Policy: [Policy requirements](https://www.nature.com/documents/nr-editorial-policy-checklist.pdf) (Download the link to your computer as a PDF.)

Furthermore, please align your manuscript with our format requirements, which are summarized on the following checklist: [Communications Earth & Environment formatting checklist](https://www.nature.com/documents/commsj-phys-style-formatting-checklist-article.pdf)

and also in our style and formatting guide [Communications Earth & Environment formatting guide](https://www.nature.com/documents/commsj-phys-style-formatting-guide-accept.pdf) .

*** DATA: Communications Earth & Environment endorses the principles of the Enabling FAIR data project (<http://www.copdess.org/enabling-fair-data-project/>). We ask authors to make the data that support their conclusions available in permanent, publically accessible data repositories. (Please contact the editor if you are unable to make your data available).

All Communications Earth & Environment manuscripts must include a section titled "Data Availability" at the end of the Methods section or main text (if no Methods). More information on this policy, is available at <http://www.nature.com/authors/policies/data/data-availability-statements-data-citations.pdf>.

If a community resource is unavailable, data can be submitted to generalist repositories such as [figshare](https://figshare.com/) or [Dryad Digital Repository](http://datadryad.org/). Please provide a unique identifier for the data (for example a DOI or a permanent URL) in the data availability statement, if possible. If the repository does not provide identifiers, we encourage authors to supply the search terms that will return the data. For data that have been obtained from publically available sources, please provide a URL and the specific data product name in the data availability statement. Data with a DOI should be further cited in the methods reference section.

REVIEWER COMMENTS:

Reviewer #1 (Remarks to the Author):

Please see the attached document

Reviewer #2 (Remarks to the Author):

Please see the attached PDF

Reviewer #3 (Remarks to the Author):

Review of "Transient ice ring observed during the 15 January 2022 eruption of Hunga volcano"

By Andrew T. Prata, Roy G. Grainger, Isabelle A. Taylor and Alyn Lambert

This study investigates a ring of cloud formed around the main volcanic plume after the underwater volcanic eruption from Hunga Tonga in 2021. Satellite retrievals show that the cloud consisted of very small (~2 μm) ice crystals and lasted for approximately 2 hours. The authors hypothesize that this ice cloud was formed as a result of the Lamb wave that reduced the pressure and led to an adiabatic temperature decrease. The authors use meteorological data together with a model for ice crystal growth to show how an ice crystal cloud could form due to this temperature decrease.

The manuscript uses advanced processing of satellite data to both investigate the ice cloud and the Lamb wave propagation. The manuscript is in general well written and structured. However, in some parts of the text the language is a bit vague and not all features in the figures are explained. The topic of the paper is well within the scope of Communications Earth & Environment. The cloud of investigation is a unique feature that deserves investigation, but it has only been observed during this volcanic eruption. The authors motivate the importance of studying this event as a natural experiment to study ice crystal growth. However, I think there could have been more discussion of these processes in the manuscript. I also have some concerns regarding uncertainty estimates for the satellite retrievals and the ice crystal modelling. I would recommend this manuscript for publication if the author can address the following concerns and comments appropriately.

General comments:

All satellite measurements include some sort of uncertainty. I think the paper needs more discussion with regards to the uncertainty of the size of the ice crystals. On page 8 you write "this process reveals a practical lower-limit for which the effective radius measurements can be retrieved using thermal infrared measurements." Being close to the limits of what the retrieval can handle surely must mean that there are substantial uncertainties in the effective radius estimates.

Figure 4a shows results of retrievals of ice crystal radius over time. The error bars show the standard deviation of the data within the region of investigation. We do not however get any uncertainty estimated of the effective radius retrievals. If the retrievals have an uncertainty of even 10 % then that will cover a quite large range of the ppmv estimates from the model shown in Figure 3a. Thus, knowing the uncertainty range is crucial for estimating model performance. There are quite large error bars (variation) for the satellite measurements and the measurements do not follow quite the same pattern as the red line. Is it really possible to deduce which ppmv is the best fit to such a high accuracy as you state in the manuscript?

There is quite some discussion of the vapor concentrations needed for the formation of the ice crystals. An important aspect of the ice crystal formation that is not discussed at all is the initial size of the ice crystals before they start growing. The manuscript simply states that the authors assume an initial particle radius of 0.01 μm . Can you justify this starting size? What type of particles of this size would be present in the upper troposphere? The size of the initial seeds will have a large impact on the growth of the crystals and the fit of the crystal sizes to the observations. I think the choice of this size needs a more thorough motivation. It would also be interesting to perform a sensitivity analysis of the initial particle size and its importance for the growth over time. Using larger particles would most likely require less moisture to reproduce the growth in the observations. I think the manuscript would also need a discussion of what particles that could be present to initiate heterogeneous ice nucleation and if enough particles could be present to form such a large ice cloud. A discussion if homogeneous ice formation could be responsible for such a cloud would also be suitable in this manuscript.

The authors' model for ice crystal growth requires more vapor present at 100 hPa than the ERA data show at this level. The authors speculate that the UTLS water vapor was enhanced by the 13 January explosion (2 days prior to the formation of the cloud). However, atmospheric transport in the UTLS in this region is impacted by the subtropical jet and other high-level winds. Water vapor and particles emitted in the UTLS will be transported eastward and would most likely not be present in the region over Hunga Tonga 48 hours later. Have you taken this into consideration?

Specific comments:

Page 1 line 036: "Record breaking heights have been reported...". Heights of what? This is vague.

Page 2 line 083: "only modest amounts of SO₂ were measured" Was this in the stratosphere or in the atmosphere in general?

Page 3 line 101-103. It would have been very nice to see the evolution of the ice ring in satellite data images to give the

reader a better view of the development. Maybe such a figure could be included in a supplementary.

Page 3 lines 111-114. You claim that the very large difference in BTM between the 8.6 μm and 11.2 μm indicates small crystal sizes. It would be nice to have an explanation of why.

Page 3 lines 120 - 121. I agree that ice crystals of 2 μm are very small. However, calling them extremely small and then in the next part of the sentence write that tropical tropopause cirrus clouds are 2.5 to 25 μm seems a bit contradictory. There is not a very large difference between 2 and 2.5 μm .

Page 3 lines 133-136: It would have been very nice to show these ice mass fluctuations in a figure.

Page 5 lines 189-190. You write that the mean temperature reduction is 6 K and that the maximum reduction is 8 K. However, the temperature scale in the figure is between -5 and +5 K. Thus these results cannot be seen in the figure. Please change the color scale.

Page 5 lines 190 to 192: This sentence is confusing. We have no prior information about a ozone weighting function or what is used for. Please consider rewriting this sentence.

Fig 2a line 215: It is confusing that you have rings at different distance from the volcano in this figure compared to Figure 1 and 3. It would be easier for the reader if you kept the solid circle marking the same area in this figure as in Figure 1 and 3 and show the other features related to the Lamb wave with a differently formatted ring.

Fig 2 line 220: What is RTTOV? This acronym is not explained or used in any other part of the paper.

Page 6 line 254-256: This sentence is confusing. Please consider rewriting it.

Page 7 line 309: You use the word model in quite a lot of contexts in the manuscript. It would be good to specify here which of the models you are referring to.

Page 7 line 313: The ERA data showed 4-5 ppmv but here you instead use 5-6 ppmv without explaining why.

Page 7 line 317: "can mostly explain" What do you mean by mostly here? Are there other factors that can explain this? The expression is vague and a bit confusing.

Fig 4b line 398: What are the grey dashed lines shown in this figure? Where does the data for the red and blue bars come from?

Page 13 line 583: Do you model the growth of one ice crystal or many. In case you model many, in what volume of air do you then model them?

Communications Earth & Environment is committed to improving transparency in authorship. As part of our efforts in this direction, we are now requesting that all authors identified as 'corresponding author' create and link their Open Researcher and Contributor Identifier (ORCID) with their account on the Manuscript Tracking System prior to acceptance. ORCID helps the scientific community achieve unambiguous attribution of all scholarly contributions. You can create and link your ORCID from the home page of the Manuscript Tracking System by clicking on 'Modify my Springer Nature account' and following the instructions in the link below. Please also inform all co-authors that they can add their ORCIDs to their accounts and that they must do so prior to acceptance.

Version 1:

Decision Letter:

Dear Dr Prata,

Your manuscript titled "Transient ice ring observed during the 15 January 2022 eruption of Hunga volcano" has now been seen by Reviewer #2, whose comments appear below. In light of their advice we are delighted to say that we are happy, in principle, to publish a suitably revised version in Communications Earth & Environment.

We therefore invite you to revise your paper one last time to address the remaining concerns of our reviewer. At the same time we ask that you edit your manuscript to comply with our format requirements and to maximise the accessibility and therefore the impact of your work.

EDITORIAL REQUESTS:

****Please take care to match our formatting and policy requirements. We will check revised manuscript and return manuscripts that do not comply. Such requests will lead to delays. ****

SUBMISSION INFORMATION:

OPEN ACCESS:

Communications Earth & Environment is a fully open access journal. Articles are made freely accessible on publication. For further information about article processing charges, open access funding, and advice and support from Nature Portfolio, please visit <https://www.nature.com/commsenv/open-access>

Link Redacted

Best regards,

Joe Aslin

Deputy Editor,
Communications Earth & Environment

Consulting Editor,
Communications Sustainability

<https://www.nature.com/commsenv/>
Twitter: @CommsEarth

REVIEWERS' COMMENTS:

Reviewer #2 (Remarks to the Author):

Dear Andrew Prata and Co-Authors, congratulations on this excellent article. In my opinion, it is now scientifically sound and ready for publication. I have a few minor comments (see the report) that I would recommend including in the final version.

** Visit Nature Portfolio's author and referees' website at www.nature.com/authors for information about policies, services and author benefits**

Review of Transient ice ring observed during the 15 January 2022 eruption of Hunga volcano

The manuscript describes satellite observations of an expanding ring of extremely fine ice crystals which formed in the upper troposphere/lower stratosphere in response to the 15 January 2022 Hunga eruption, Tonga. This is a uniquely observed atmospheric phenomenon. By combining radiometric satellite observations with previously published observations of the atmospheric waves generated by the eruption, the authors hypothesise that the ice ring formed due to the passage of the generated Lamb wave. Finally, the authors support this hypothesis by developing a model of ice crystal growth in response to the Lamb wave, showing agreement between the predicted ice crystal sizes and their timescale of formation.

As has been reported in many published articles, the 15 January 2022 Hunga eruption produced many phenomena unique within the period of modern observations. The ice ring described in this manuscript has not been previously described before. That the authors can support this observation with a quantitative model of formation is a strength of the paper and makes this a well-rounded study. Reading the paper, it became apparent that the study is more about atmospheric processes triggered by the eruption, rather than the eruption itself. Therefore, I think that the manuscript is of more specific interest to atmospheric scientists rather than volcanologists. However, given the uniqueness of the eruption, I think the paper will be of general interest to a wider audience of Earth and atmospheric scientists.

I have no serious concerns about the paper. My biggest issue is that I think there are a couple of areas where the observations and methods can be made a little clearer to the readers. I detail these below but, as an example, it is not visually clear from Fig 1. that the ice ring is a) a ring at all rather than a disc and b) propagating outwards (see major comment 3). Both of these could be addressed by showing a timeseries of satellite images. A second issue is that I think the authors need to be careful with how they compare their modelled results to the observations (see major comments 5 and 6).

To conclude, I think this is a good paper which does an excellent job of interrogating a novel observation, presenting a quantitative model to explain it. There are some minor areas where the manuscript can be improved, as summarised in the above paragraph and detailed below. In the following, I provide some detailed comments, separated into those on the scientific content, and those on the manuscript presentation and style. I want to note that I only provide comments on the presentation and style to help the authors as they revise the manuscript. I have also provided a short list of references at the end of this review of papers which I refer to in the comments below. Whilst I think all of these may be useful for the authors as they revise the manuscript, I want to make clear I am not asking for them to be added as references to the paper.

We thank Reviewer 1 (R1) for their detailed review. We agree with the majority of comments and have implemented the changes as suggested (see responses below). On the point of interest to volcanologists, we agree that the major phenomenon studied here falls under atmospheric science (in particular, cloud microphysics); however, one of the main motivations for publishing this study in an Earth science journal is to show that this particular feature of the Hunga eruption cloud should not be confused with an umbrella plume, which is of particular importance to volcanologists attempting to model the plume dynamics and comparing with satellite observations. All references cited within this response are provided at the end of this document.

Major comments

1. Lines 60-61. This may be ignorance on my part, but why do umbrella clouds and overshooting tops indicate that a volcanic plume reached the tropopause? My understanding is that these will form if 1) a plume is able to reach a neutral buoyancy level and 2) the plume is strong, i.e., the ambient wind is insufficiently strong to significantly impact the plume rise. Is it not possible for both of these conditions to be true for a plume which does not reach the tropopause?

Thank you for making this point. R1 is correct. We wanted to highlight an observation where it can be deduced that material can enter the stratosphere. We have clarified and revised the statement as follows:

“The observation of an umbrella cloud indicates that a volcanic plume has reached its level of neutral buoyancy (Holasek et al. 1996; Pavolonis et al., 2018; Prata et al. 2020). Volcanic plumes with a warm centre relative to the surrounding umbrella can indicate that material has entered the stratosphere (Lucas, 2023)”.

2. Lines 74-75. In fact, eruptive activity commenced on 19 December 2021, and not 13 January 2022 (Gupta et al., 2022).

Modern records of eruptive activity at Hunga volcano have been made in 1988, 2009, 2014, 2021, 2022 etc (see GVP database entry for Hunga Tonga-Hunga Ha'apai). The eruptive activity that commenced on 13 January 2022 at around 15:37 UTC is of relevance here because we hypothesise that the 13 January eruption provided both a source of water vapour and INPs in the UTLs for the ice ring to form. We have adjusted the sentence (deleting the term "began") to avoid any confusion or ambiguity regarding eruptive history. The amended sentence is:

"On 13 January 2022 at around 15:37 UTC, significant phreatomagmatic activity at Hunga volcano (Kingdom of Tonga) was observed by the Advanced Himawari Imager (AHI) aboard the Himawari-8 satellite."

3. Lines 88-91 and Figure 1. After reading this description of the ice ring, it took me a little while to work out that the ice ring is actually the grey, diffuse cloud beneath the uppermost umbrella in Figure 1a. I think this is because, to me, it does not look very ring-like. Instead, to me, it looks like a disc-shaped umbrella cloud, with the inner portion visually obscured by the higher umbrella. In fact, from the single image shown in Figure 1a alone, it is impossible for the reader to identify this cloud as a ring with an inner and outer diameter, rather than a disc which is continuous to the vent location. To help the reader here, it may help to show a timeseries of true colour images (and maybe the other derived images as well), from which the reader will be able to identify the ring shape and see the expansion.

Based on the single image in Fig. 1a, we agree that it is impossible to conclude that the shape is definitively a ring but is also impossible to infer that it is a disk. Both shapes require an assumption of the structure of the cloud over the vent (which, as R1 acknowledges, is obscuring the view at this stage). In our initial submission we provided indirect evidence of the ring shape (i.e. Figure 2(a) of the original manuscript), where we see a clear ring shape in the lower-middle stratosphere temperature perturbations (derived from thermal ozone and window channel measurements). From a theoretical standpoint, a radially outward propagating compressional wave would not produce a continuous disk. Rather, a compressional wave will produce radial peaks and troughs in temperature, which is what we propose and model in our study. To help the reader and respond to this comment, we have added further detail on the initial description of the cloud as follows:

"From 04:47-05:17 UTC, after the dome collapsed, a grey diffuse cloud, surrounding the main eruption column, formed below the uppermost umbrella. The diffuse cloud had a striking circular shape, which can be appreciated by comparing the edge of the cloud with the 300 km circle radius annotated on the figure panels of Fig. 1, which show the Himawari satellite's view of the eruption at 05:17 UTC. Due to the presence of the main plume over the volcano, at this time (05:17 UTC), the shape of the diffuse cloud could be interpreted as a 'disk' or a 'ring' centred over the volcano."

We have now added an animation (Supplementary Movie 1) that shows the full observation period of ice detection from Himawari over the volcano from 13-15 January 2022. This movie includes imagery at 05:37 UTC on 15 January where the ring shape of the ice cloud can be identified. In our revision we comment on this as follows:

“However, if the Himawari data are animated (Supplementary Movie 2), an ice cloud signature in the shape of a ring can be identified (in particular at 05:37 UTC on 15 January), and so we refer to it hereafter as an ‘ice ring’.”

We have also added a new figure (Fig. 3 of the revised manuscript) describing the ice ring’s development and have added annotations. The new figure and animation serve two purposes:

- 1) To provide a clearer visual representation of the phenomenon we are describing.
- 2) To provide evidence that there was in fact an ice cloud layer over the volcano at the time of the ~04:07 UTC 15 January eruption (in line with a response to a comment from R3).

4. Line 196 and Figure 2a. You refer to a Lamb wave initiated at 04:28 UTC, as described by Wright et al. (2022). Inspection of Wright et al.’s Extended Data Fig 1e shows that this origin time was determined by recording the time of peak pressure disturbance at ground level stations, and performing a linear regression back to Hunga volcano. Jarvis et al. (2023) repeat this for barometers in New Zealand, finding a comparable result. However, they also note that, at each station prior to the peak pressure, there is a gradual increase in pressure over a period of approximately 15 minutes. Thus, if you actually perform the linear regression using the times at which the Lamb wave onset is detected, rather than the peak, an origin time of 04:15 is retrieved. This is consistent with the timing of the primary seismoacoustic event which is recorded (Matoza et al., 2022). I therefore ask the authors if it would be more appropriate to use a Lamb wave origin time of 04:15. Alternatively, refer to the circle in Fig 2a as the location of the peak of the Lamb wave, rather than the leading edge.

Thank you for raising this point. It is unclear which origin time is more appropriate to use here as it is a topic of active research. Therefore, as suggested by the reviewer, we have explicitly stated that the circle corresponds to an origin time based on the peak pressure signal as follows:

“Figure 2a shows how the temperature response in the lower-middle stratosphere compares with present estimates of the Lamb (red circle) and Pekeris (blue circle) wave phase speeds and an assumed origin time of 04:29:30 UTC which has been derived from the timing and location of the peak pressure signal in surface pressure stations (Wright et al., 2022; Jarvis et al., 2024).” Note we have also added a circle representing the Pekeris wave which should also be considered as influencing UTLS temperatures.

5. Lines 320-324. The authors state that they are able to reproduce the observed particle radii oscillating in size with time. However, the evidence for this oscillation is not strong; based on the error bars shown in Fig 4a, the amplitude of this oscillation is smaller than the uncertainty on the measurements. Additionally, the modelled temporal oscillation does not appear to be in-phase with the observed oscillation (if it is there). The authors also state that the best agreement between modelled and observed results occurs for an assumed water vapour concentration of 5.45 ppmv. Can the authors quantify this level of agreement and provide an estimate of what range of concentrations still provide consistent results.

We agree that the evidence for the oscillations in the presented timeseries is not strong and have acknowledged this in our revised statement as follows:

“In addition, including the temperature wave perturbation reproduces what appears to be oscillations in the satellite-retrieved effective radius with time; however, on the basis of the present analysis, this inference is speculative, as the standard deviations of the satellite-retrieved effective radii are larger than the variations of mean effective radii with time.”

Note that the error bars do not represent the uncertainty on the satellite measurements. They are a measure of the variation in the distribution of the size of particles contained in the ice ring because we are taking the standard deviation of the retrieved particle size of a distribution of pixels within a spatial area at each observation time. In response to this reviewer comment and comments from R2 and R3, we have completely revised and improved our ice particle modelling approach. The major changes are:

1. We now model a size distribution of ice particles, rather than a single ice particle.
2. We now provide ensemble simulations and plot the mean and standard deviation of the effective radius derived from the ensemble simulations, making the observations and model simulations directly comparable.
3. We set the water vapour concentration based on varying levels of ice supersaturation now because we now argue that the atmosphere was supersaturated due to the presence of ice clouds over the volcano before the 15 January eruption.

In our new simulations the modelled particle size does appear more in phase with the satellite measurements though the agreement is not perfect. We might expect this due to other reasons not accounted for in our model such as wind advection of the ice clouds within the spatial region.

6. Fig. 4a. Why does the modelling shown here use 04:47 as an origin time? This is the time of the first Himawari image which observed the ice ring, so presumably ice crystal growth must have started earlier than this? How sensitive is the result of 5.45 ppmv as the best fitting vapour concentration to the choice of origin time.

Recall that the modelling is compared to satellite retrievals corresponding to a region enclosed by two concentric circles corresponding to radial distances of 250 km and 300 km. Within this specific region, the first ice particles are detected at 05:07 UTC (ice particles are observed at 04:47 but these retrievals do not fall within the spatial region we selected for further analysis). The modelling uses 04:47 UTC (20 minutes before ice particles start to form) as a start time because this is the time we find that the cold phase temperature perturbation in the UTLS begins (i.e. the large decrease in ambient temperature from 04:47 to 04:57 UTC in Fig. 4 of the revised manuscript).

Our new modelling results show the sensitivity of the ice particle modelling to water vapour concentration by presenting ensemble modelling simulations at 4 different levels of ice supersaturation (see Fig. 5 of the revised manuscript).

Minor comments

Scientific content

1. Line 44. Can you quantify the adiabatic decrease in ambient temperature?

In our ensemble simulations we now consider a range of ambient temperature reductions from -2 to -0.4 K, which are reported in Sect. 2.5 of the revised manuscript. For interest, the previous temperature reduction we used in our single particle simulations was -1.3 K.

2. Line 59. I suggest adding the word “explosive” in front of “volcanic eruptions”.

Done.

3. Lines 175-176. The sentence “Examples of such impulses ...” requires references. I’m particularly surprised that atmospheric Lamb waves have been observed to be generated by earthquakes.

References added.

4. Line 189. The text says that the Lamb wave caused a maximum temperature reduction of 8 K in the lower stratosphere. But this magnitude of decrease goes beyond the colour bar scale in Figure 2a, which only reaches -4 K. Surely the colour bar in Figure 2a should be extended to this.

Thanks for pointing this out. We have double checked these values and the maximum reduction is actually 8.7 K. We revised this value in the text and the colour bar has been amended to range from -9 K to +9 K.

5. Lines 190-193 and Figure 2b. The sentence “We estimate that these ...” and Figure 2b required me to do a bit of further reading to understand what is described here. I’m personally not familiar with weighting functions. However, from the context here and in the rest of the paper, I think they characterise the contribution to the signal in different wavelength bands from different parts of the Earth’s atmosphere. Some additional explanation and/or references might be needed for general readers who are not remote sensing scientists.

We have added a clarifying sentence on weighting functions for the non-expert and provided a reference to the definitive textbook on atmospheric sounding i.e. Rodgers (2000). We have also added annotations to Fig. 2(b) to help the reader understand the use of weighting functions here. The clarifying sentence is:

“The weighting function is defined as the derivative of the transmittance profile with respect to the vertical coordinate (e.g. height or pressure) for a given wavelength or wavelength interval for a broadband infrared channel (Rodgers, 2000). Weighting functions therefore represent the relative contributions each atmospheric level makes to the measured top-of-atmosphere radiance in a particular channel.”

6. Lines 261-263. The authors state that the temperature perturbation caused by the Lamb wave at pressures of 100 hPa is predicted to be ~ 1.3 K and claim that this is sufficient to trigger ice nucleation. However, on line 249, the authors state that the perturbation needs to be at least 1-2 K. So, this inferred perturbation is only just large enough. It’s probably worth noting this.

Thank you for this comment. This comment in addition to other reviewer comments led us to revise our analysis in light of the strong evidence that the atmosphere was supersaturated with respect to ice prior to the 15 January eruption. We now account for this scenario in our new modelling approach. In addition, in light of the Watanabe et al. (2022) study, which we were not aware of at the time of writing the original manuscript, we no longer attribute the Lamb wave to being the sole cause of the UTLS temperature perturbation and have removed the analysis which attempts to precisely quantify the temperature perturbation at a given pressure level based on the method presented in Otsuka (2022). We now propose that it’s possible that a Pekeris wave influenced the UTLS temperature change. Though this is still an active area of research. To account for this uncertainty, we now provide ensemble simulations which account for a range of UTLS temperature reductions and leave the attribution of the temperature changes in the UTLS as an open research question.

7. Line 435. How has the value of 0.111 for the effective variance been chosen?

The value for effective variance and choice of the size distribution used in the present study follows that of the Moderate-Resolution Imaging Spectroradiometer (MODIS) cloud retrieval algorithm collection 6, which has been informed by radiative transfer model sensitivity studies and numerous field campaigns (see Platnick et al., 2017, for details). We have added this statement to the revised manuscript.

Presentation and style

1. Figure 1a. The white font makes it very difficult to read the label '300 km'.

We have added a white background to the label and used a black font to make it clear to the reader.

2. Fig. 1. The caption needs to describe what the circle at 300 km corresponds to.

Description added.

3. Throughout the manuscript, the vertical structure of the atmosphere is described in terms of pressure. Whilst this makes sense from an atmospheric perspective, it would help the more general reader if reference were also made to the altitudes being considered. For example, when referring to the pressure range 6-52 hPa, or to 100 hPa, it would be good to also state the corresponding altitudes.

Corresponding altitudes (in km) have been added where appropriate to help readers that are not familiar with pressure altitudes.

4. Fig. 4. Please define AHI in the caption, as it isn't defined in the text until the methods section. Also, the γ parameter shown in the legend are missing units whilst the symbol γ is used in the caption but the symbol ζ is used in the text. It might also be useful to add a reference to Equation 28 in the caption, to direct the reader to where the damped oscillator model is defined.

In our revised manuscript we define the Advanced Himawari Imager (AHI) in the second paragraph of the Introduction section. We also explicitly spell it out in the revised Fig. 1 caption and again in the Methods section. Thank you for noting the typos in Fig. 4. We have corrected the Greek symbols in our revised text and the revised version of Fig. 4. We haven't added a reference to the equation for a damped harmonic oscillator as it is a standard and widely known mathematical function in physics. Any reader can find it on Wikipedia or in a physics textbook.

5. Line 414. Please state what the "effective radius" refers to. Is it ice crystals specifically, or can it be any particulate matter?

Done. Effective radius refers to ice particles in the present study but it is a general term that can be used for any distribution of particles (e.g. volcanic ash, liquid water droplets, sulphate aerosols etc).

6. Line 426. Add the word "size" between "gamma" and "distribution"

Done.

7. Line 426. Add the word "index" between "refractive" and "given"

Done.

8. Line 527. Add the word "perturbation" between "temperature" and "time-series" to make it clear that the quantity given by equation 9 is the temperature perturbation, rather than the absolute temperature.

Done.

9. Line 541. Maybe add a reference to Fig. 3 after stating that “ $T_{\text{ambient}} = 193 \text{ K}$ at 100 hPa” to remind the reader where this value comes from.

Accepted but this revision is no longer needed as we have revised the modelling simulation approach. All model initial conditions and variations are reported in text. See Sect. 2.5 of the revised manuscript.

References

Jarvis, P. A., Caldwell, T. G., Noble, C., Ogawa, Y. & Vagasky, C. (2023). Volcanic lightning reveals umbrella cloud dynamics of the January 2022 Hunga Tonga-Hunga Ha’apai eruption. EarthArXiv. <https://doi.org/10.31223/X5D09J>

Matoza, R. S., Fee, D., Assink, J., Iezzi, A. M., Green, D. N., Kim, K. et al. (2022). Atmospheric waves and global seismoacoustic observations of the January 2022 Hunga eruption, Tonga. *Science*, 377, 95-100. <https://doi.org/10.1126/science.abo7063>

References

Holasek, R. E., Self, S., and Woods, A. W.: Satellite observations and interpretation of the 1991 Mount Pinatubo eruption plumes, *Journal of Geophysical Research: Solid Earth*, 101, 27635–27655, <https://doi.org/10.1029/96JB01179>, 1996.

Jarvis, P. A., Caldwell, T. G., Noble, C., Ogawa, Y., and Vagasky, C.: Volcanic lightning reveals umbrella cloud dynamics of the January 2022 Hunga Tonga-Hunga Ha’apai eruption, <https://doi.org/10.5281/ZENODO.8011902>, 2023.

Lucas, C.: Determining the height of deep volcanic eruptions over the tropical western Pacific with Himawari-8, *J. South. Hemisph. Earth Syst. Sci.*, 73, 102–115, <https://doi.org/10.1071/ES22033>, 2023.

Otsuka, S.: Visualizing Lamb Waves From a Volcanic Eruption Using Meteorological Satellite Himawari-8, *Geophysical Research Letters*, 49, <https://doi.org/10.1029/2022GL098324>, 2022.

Pavolonis, M. J., Sieglaff, J., and Cintineo, J.: Automated Detection of Explosive Volcanic Eruptions Using Satellite-Derived Cloud Vertical Growth Rates, *Earth and Space Science*, 5, 903–928, <https://doi.org/10.1029/2018EA000410>, 2018.

Platnick, S., Meyer, K. G., King, M. D., Wind, G., Amarasinghe, N., Marchant, B., Arnold, G. T., Zhang, Z., Hubanks, P. A., Holz, R. E., Yang, P., Ridgway, W. L., and Riedi, J.: The MODIS Cloud Optical and Microphysical Products: Collection 6 Updates and Examples From Terra and Aqua, *IEEE Transactions on Geoscience and Remote Sensing*, 55, 502–525, <https://doi.org/10.1109/TGRS.2016.2610522>, 2017.

Prata, A. T., Folch, A., Prata, A. J., Biondi, R., Brenot, H., Cimarelli, C., Corradini, S., Lapierre, J., and Costa, A.: Anak Krakatau triggers volcanic freezer in the upper troposphere, *Scientific Reports*, 10, <https://doi.org/10.1038/s41598-020-60465-w>, 2020.

Prata, A. T., Grainger, R. G., Taylor, I. A., Povey, A. C., Proud, S. R., and Poulsen, C. A.: Uncertainty-bounded estimates of ash cloud properties using the ORAC algorithm: application to the 2019 Raikoke eruption, *Atmos. Meas. Tech.*, 15, 5985–6010, <https://doi.org/10.5194/amt-15-5985-2022>, 2022.

Rodgers, C. D.: *Inverse methods for atmospheric sounding : theory and practice*, World Scientific, Singapore, 2000.

Watanabe, S., Hamilton, K., Sakazaki, T., and Nakano, M.: First Detection of the Pekeris Internal Global Atmospheric Resonance: Evidence from the 2022 Tonga Eruption and from Global Reanalysis Data, *Journal of the Atmospheric Sciences*, 79, 3027–3043, <https://doi.org/10.1175/JAS-D-22-0078.1>, 2022.

Wright, C. J., Hindley, N. P., Alexander, M. J., Barlow, M., Hoffmann, L., Mitchell, C. N., Prata, F., Bouillon, M., Carstens, J., Clerbaux, C., Osprey, S. M., Powell, N., Randall, C. E., and Yue, J.: Surface-to-space atmospheric waves from Hunga Tonga–Hunga Ha’apai eruption, *Nature*, 609, 741–746, <https://doi.org/10.1038/s41586-022-05012-5>, 2022.

Transient ice ring observed during the 15 January 2022 eruption of Hunga volcano
Andrew T. Prata¹, Roy G. Grainger, Isabelle A. Taylor and Alyn Lambert

General:

The formation mechanisms of a transient ice ring consisting of $\sim 2 \mu\text{m}$ ice crystals at an altitude of about 14-17 km observed by satellite during the eruption of the Hunga Tonga volcano is investigated in this study. It is shown that ice formation was initiated by a temperature drop caused by an atmospheric Lamb wave triggered by the eruption. From the analysis of the environmental conditions during this temperature drop and simple calculations of ice growth the authors conclude on the ice formation mechanism, the time scale of the ice formation and growth and the final size of ice particles.

I find this work exciting because, as the authors state, the observed phenomenon can serve as an important natural experiment that helps to understand ice formation. The analyses of the environmental conditions from the satellite observations seem to me (as someone who is only marginally familiar with these methods) to be carefully done. Also, the article is well structured and can be read fluently.

I was asked to review the article because of my expertise in the area of ice cloud formation and evolution, so I will focus my further report on that aspect of the manuscript.

Unfortunately, I have to say that the analysis and interpretation of the ice formation and evolution does not seem mature to me. For example, it was not discussed that there are two possible ways for ice to form, but the ice ring was simply assigned to one mechanism. Further, the processes that influence the growth of ice crystals have not been fully discussed and taken into account in the analyses. In my opinion, it needs to be shown that the simplified ice modeling leads to realistic results. The calculations themselves should be performed for the respective observed ranges of the parameters involved - and not just for a specific value - to show that the result, namely the $2 \mu\text{m}$ size of the ice crystals, is robust. More detail is given in the specific comments listed below in the order of their appearance in the manuscript.

Overall, I unfortunately cannot recommend the manuscript for publication in its current form, although the material definitely has the potential for an exciting study. I would therefore like to encourage the authors to revise and resubmit the article.

We thank Reviewer 2 (R2) for their detailed review and in particular on the ice cloud microphysics. We have made great efforts to improve the ice particle modelling to address R2's concerns. The results from our revised analysis, in particular modelling a size distribution (and time varying saturation ratio), lead to better agreement with the satellite retrievals as will be discussed in our responses below. All references cited within this response are provided at the end of this document.

Specific comments:

1) line 120: 'The mean ice particle sizes were extremely small ($2 \mu\text{m}$ ice; $\sim 2 \mu\text{m}$; Fig 1d) when compared to tropical tropopause cirrus clouds [$2.5\text{--}25 \mu\text{m}$ ice radius; 20].'

The size of ice crystals depend on the stage of their development. All in-situ formed ice particles appear at very small sizes, because they form either heterogeneously on INPs or homogeneously on supercooled solution particles and thus have initially about the size of the particles, which is $< 1 \mu\text{m}$ ice. In the course of the evolution of the ice cloud, they grow as long as the environment is supersaturated. The growth depends on the number of ice crystals, since the ice crystals compete for the available water vapor. At the cold temperatures and low water vapor mixing ratios prevailing in the ice ring, the ice particles cannot grow to mean sizes substantially larger than $2 \mu\text{m}$ ice, $\sim 2 \mu\text{m}$; especially since the time for growth is short (see your line line 120: 'the whole process (from formation to dissipation) took 2 h to complete'). You show $\sim 2 \mu\text{m}$; this in Figure 4 (the cooling / growth phase is only about 10min). So the reason that in most cases the ice particles grow larger under natural TTL conditions is that the growth periods are longer, or that the ice crystal concentration is very small. A typical mean ice crystal size in the TTL is $\sim 8 \mu\text{m}$ ice (Krämer et al., 2020, ACP).

- I recommend to discuss in the paper that a shorter growth periods and/or a larger ice crystal concentration is likely the reason for the smaller ice particles observed in the ice ring in comparison to natural TTL ice clouds.

- Also, I recommend not saying that the ice crystals are extremely small, but that such small ice crystals are unlikely to occur in the atmosphere under natural conditions.

Thank you for these comments and suggestions. We appreciate these insights and have incorporated them into our revised manuscript. We couldn't find a statement or discussion in Krämer et al. (2020) stating that "typical mean ice crystal size in the TTL is $8 \mu\text{m}$ ". Looking at their figures (see Figure S2 for R_{ice}), it looks like the median at 15-18 km is colour-coded according to the 6-18 μm radius range. We have revised the statement to make mention of both factors discussed (namely, ice crystal concentration and growth duration) and have deleted any mention of "extremely small" ice particles.

The relevant revised text is:

"The mean ice particle effective radii within the ice ring at 05:17~UTC was $\sim 2 \mu\text{m}$ (Fig. 1d). Tropical tropopause cirrus cloud sizes have been reported in the range from 2.5-25 μm radius (Woods et al., 2018); however, the size observed depends upon the stage of cloud development. The Hunga ice ring particle sizes would fall into the smallest, median R_{ice} size range (1-6 μm) reported by Krämer et al. (2016, 2020) who compiled a cirrus ice particle size climatology from in situ aircraft data. Krämer et al. (2020) show that the median ice particle sizes for cirrus clouds at similar altitudes (~ 15 -18 km) lie in the range from 6-18 μm ."

We have also added a footnote to explain how R_{ice} is defined.

2) line 123 ff: '.. ; however, the ice particles studied here are of an order of magnitude smaller, indicating a new ice nucleation pathway not previously documented for volcanic clouds.'

I think this statement is somewhat exaggerated - the observations can certainly be explained by conventional ice formation mechanisms - see the previous comment (on line 120). In my opinion, the difference to previous volcano-induced ice clouds lies in the short period of the Lamb wave, and possibly in the number of ice crystals (see also previous comment). Is there any information about this? That would be helpful, and it would also be advisable to estimate the time scales for ice growth during previous volcanic eruptions. Nevertheless, I find the observation and description of this phenomenon to be important.

We were unable to find published estimates of the timescales (e.g. in seconds or minutes) for ice particle growth in volcano-induced ice clouds. Durant et al. (2008), which we cite, provides a comprehensive overview of volcano-induced ice clouds. We agree with the reviewer that a possible reason for the small ice particle sizes is due to the short duration of ice supersaturation triggered by the atmospheric waves caused by the eruption. In response to this reviewer comment we have revised and added the following text:

“Ice particle sizes of $\sim 20 \mu\text{m}$ associated with volcanic eruptions have been observed before (Durant et al., 2008; Prata et al., 2020) and the particle sizes observed have been attributed to overseeding (i.e. high number concentrations of ice nucleating particles; Durant et al., 2008); however, the ice particles studied here are of an order of magnitude smaller, indicating an earlier stage of ice particle growth that has not previously been documented for ice clouds associated with volcanic eruptions. Such small ice particles are unlikely to be observed under natural conditions. In theory, the smallest ice particles will be observed at the very beginning of the ice particle growth phase during cloud development. Once nucleated, ice particles will initially grow via vapour deposition. Particle growth via vapour deposition is principally driven by the level of supersaturation, initial (dry) particle size distribution and the ice nucleating particle (INP) number concentration (Mason, 1975; Rogers and Yau, 1989, Pruppacher and Klett, 2010; Lohmann et al., 2016). The longer an environment remains supersaturated with respect to ice, the larger an ice particle will be expected to grow. Thus if the environment is subject to only a short period of supersaturation then small ice particles may be expected. For distributions of ice particles, low number concentrations will lead to larger particle sizes (and vice versa) as individual ice particles compete for the available background ambient water vapour.”

3) line 228ff: ' For ice to form directly from water vapour, ice nucleating particles (INPs) must be present.' Ice particles can also form on supercooled solution aerosol particles, see my comment 1). The two pathways of ice formation differ in the supersaturation threshold that needs to be reached. For ice nucleation on INPs (heterogeneous) the threshold is lower ($\sim 110\text{-}120\%$ RH_{ice} - relative humidity with respect to ice) than for freezing of supercooled solution aerosols (homogeneous, $\sim 150\text{-}160\%$ RH_{ice} at $T \sim 192\text{K}$ and 100 hPa). Is it possible to retrieve the ambient RH_{ice} from the observations ? That would be helpful to learn about the freezing possible mechanism. In any case, though heterogeneous freezing seems likely in a plume of a volcano because volcanic ash particles are known to be efficient INPs (e.g. Rolf et al., 2012, ACP), the two ice formation pathways need to be mentioned. We now also discuss both ice formation pathways - please see response to (4) below where we have added reference to Rolf et al. (2012). We also use the analysis presented in Rolf et al. (2012) to inform our modelling of a size distribution of particles.

While it is possible to derive RH_{ice} from the ERA5 data, we realised, upon revision, that the ice clouds produced by the 13-14 January eruptive activity (prior to the 15 January eruption), provide a much more convincing explanation for both the source of supersaturated conditions and INPs for the ice ring to form. The ice clouds prior to 15 January do not appear in the ERA5 reanalysis and so RH_{ice} calculated from ERA5 does not appear to be relevant for the ice particle model simulations. We have therefore decided to remove Fig. 3 of the original manuscript (i.e. the ERA5 maps).

We now provide more detailed evidence of the source of the pre-existing ice clouds over the volcano on 15 January in Supplementary Movie 1. In our revision, we argue that the process we are seeing is a process of sublimation of pre-existing ice clouds and subsequent re-nucleation (see new Fig. 3 in the revised manuscript). We believe the observational evidence presented in our revision supports this claim.

4) line 229ff: 'It is common in volcanic eruptions for water vapour in the atmosphere to be entrained into the plume, condense to liquid water and freeze in updrafts during transport to plume-top.'

Freezing of liquid drops is an additional way to form ice clouds in comparison to the two ways explained above (see e.g. Luebke et al., 2016, ACP). It is mostly triggered by INPs (thus heterogeneous), but liquid cloud drops would for sure be larger than 2 μm ice! Plus, once frozen, the ice particles quickly grow to much larger sizes. In case of high updrafts, liquid drops can survive until they reach the temperature of $\sim -40^\circ\text{C}$ and then freeze homogeneously - but even then the sizes would be around 10-20 μm ice. So I don't think that this pathway is possible here ... Maybe this was the way the ice particles have formed in earlier volcanic eruptions? (see also comment 3)

- I recommend to consider and explain in more detail the different ways of ice formation, as mentioned above.

To address both comments (3) and (4) we have revised Section 2.3 by:

1. Mentioning both pathways of ice formation (heterogeneous INPs and supercooled solution aerosol particles).
2. Mentioning the Rolf et al. (2012) study which shows how volcanic particles are efficient INPs.
3. Adding further discussion on ice cloud (cirrus) formation drawing on studies from Luebke et al. (2016), Krämer et al. (2016, 2020), Lawson et al. (2019) and Kärcher et al. (2022). Our revised text is as follows:

"In order for ice clouds to form in the atmosphere there needs to be a source of liquid water or water vapour that is brought to supersaturation with respect to ice. For cirrus, recent work suggests that these clouds can be grouped into two broad categories: in situ origin cirrus and liquid origin cirrus (Luebke et al., 2016; Krämer et al., 2016, 2020). In situ origin cirrus clouds may form heterogeneously on ice nucleating particles (INPs) or may form homogeneously on supercooled solution aerosols (Kärcher et al., 2022). Liquid origin cirrus clouds typically contain larger ice particles than in situ origin cirrus as they form from liquid water drops which rapidly grow once frozen (Luebke et al., 2016; Lawson et al. 2019). Even in the case of strong updrafts, where liquid water droplets can survive until they reach -38°C and freeze homogeneously, the smallest liquid origin cirrus ice particle sizes are typically $\sim 10\text{-}20\ \mu\text{m}$ radius (Krämer et al., 2016).

Volcanic ash particles are known to be efficient INPs (Durant et al. 2008; Rolf et al. 2012) and it is common in volcanic eruptions for water vapour in the atmosphere to be entrained into the plume, condense to liquid water and freeze in updrafts during transport to plume-top (Woods et al. 1993; Tupper et al. 2009). Magmatic water, expelled during an eruption, is also a source of water for ice nucleation, meaning that volcanogenic ice can form in relatively dry atmospheres (Rose et al. 2004). A further source of water vapour can be added to the atmosphere via phreatomagmatic interactions (Prata et al. 2020). It is likely that all of the aforementioned mechanisms were at play during the Hunga eruption, which raises the question: Which ice nucleation pathway was responsible for the formation of the ice ring?"

5) line 240ff: '.. (and INPs provided by the 13 January eruption) ..'

See previous comments.

This statement has been removed in our revision. By adding the above revisions (reviewer points 3 and 4), the new Fig. 3 and Supplementary Movie 1, we now provide a more detailed discussion of the microphysics.

6) line 244, whole paragraph:

Would it be possible to estimate a range of ambient RH_{ice} from these retrieved environmental conditions (including uncertainty ranges)?

If not, I suggest to mention the two ice nucleation pathways in the paper and maybe note that the volcanic ash embedded in the plume can serve as efficient INP (see also comment 3).

We have now completely revised the paper and this paragraph has largely been removed and/or significantly revised. We now provide ensemble simulations which capture a range of uncertainty in the model simulations and we also note the importance of the Pekeris wave which may have influenced temperatures in the UTLS (Watanabe et al., 2022).

7) line 271: '.. that that ..' - one too much

Thanks - corrected.

8) Section 2.4 Ice particle modelling

In the simulations, one ice particle that grows/shrinks when the ambient temperature is below/above the frost point, i.e. RH_{ice} is above/below 100%, is considered. This is a quite simple view to the development of an ice cloud, because in reality the concentration of ice crystals influences the development of RH_{ice} and thus also strongly impacts the degree of the growth/shrinking (the more ice crystals there are, the faster RH_{ice} drops due to ice crystal growth and the smaller is the final ice crystal size).

a) So my question is, how realistic is the simulation of one ice particle ?

The short answer is: not very realistic. In response to this review comment we have completely revised and updated our ice particle modelling approach to take into account a size distribution of particles. We also have conducted an ensemble of model simulations to capture the uncertainty in our model assumptions. The new results are very interesting. In particular we now show that a size distribution of particles (rather than a single particle) results in a better fit to the satellite retrievals (which we have also improved - see revised method section 3.1). By introducing the physical process whereby the saturation ratio can change with time due to the distribution of particles competing for and modifying the background water vapour mixing ratio, the particle growth appears to be more consistent with the particle sizes derived from observations. The single particle cannot account for this. In our revised Fig. 5, we also show that the same simulations, without a temperature perturbation in the UTLS, lead to poorer agreement with the observations, which adds weight to the argument that the damped harmonic temperature oscillations influenced the ice particle size evolution.

b) In line 321ff you write:

'For a temperature reduction of 1.3 K at 100 hPa, water vapour $\sim 2 \mu\text{m}$; concentrations of 5.3 ppmv are required to grow spherical ice particles to $\sim 2 \mu\text{m}$; 2 μm icem within 20 minutes.' $\sim 2 \mu\text{m}$;

To be honest, I don't think you can estimate the ice crystal size with such accuracy based on this simple calculation. The size growth is very sensitive to the various parameters (e.g. to H₂O, as can already be seen in Fig. 2a), but also to temperature, pressure (on which the H₂O saturation mixing ratio at a certain temperature depends) and, as I said, on the number of ice particles and the temperature development. The simulations should to cover at least a range of temperature and its decrease over time as well as a pressure range to see how this affects the ice particle size. However, it would be even better to take the ice concentration into account.

We agree and have revised our analysis as described above. In the revised simulations, we conduct ensemble simulations which account for a range of initial ambient temperature from ~ -82 to -78 °C, pressure from ~ 75 to 115 hPa and height from ~ 16 to 18 km corresponding to five ERA5 levels. We also vary the amplitude of the UTLS temperature perturbation, ice mass accommodation coefficient and level of supersaturation. We should also note that we do not claim to validate our model with the satellite retrievals but use the satellite retrievals as a constraint to understand the plausible parameter space for the ice particle modelling.

References

Durant, A. J., Shaw, R. A., Rose, W. I., Mi, Y., and Ernst, G. G. J.: Ice nucleation and overseeding of ice in volcanic clouds, *Journal of Geophysical Research*, 113, <https://doi.org/10.1029/2007JD009064>, 2008.

Krämer, M., Rolf, C., Luebke, A., Afchine, A., Spelten, N., Costa, A., Meyer, J., Zöger, M., Smith, J., Herman, R. L., Buchholz, B., Ebert, V., Baumgardner, D., Borrmann, S., Klingebiel, M., and Avallone, L.: A microphysics guide to cirrus clouds – Part 1: Cirrus types, *Atmos. Chem. Phys.*, 16, 3463–3483, <https://doi.org/10.5194/acp-16-3463-2016>, 2016.

Krämer, M., Rolf, C., Spelten, N., Afchine, A., Fahey, D., Jensen, E., Khaykin, S., Kuhn, T., Lawson, P., Lykov, A., Pan, L. L., Riese, M., Rollins, A., Stroh, F., Thornberry, T., Wolf, V., Woods, S., Spichtinger, P., Quaas, J., and Sourdeval, O.: A microphysics guide to cirrus – Part 2: Climatologies of clouds and humidity from observations, *Atmos. Chem. Phys.*, 20, 12569–12608, <https://doi.org/10.5194/acp-20-12569-2020>, 2020.

Lohmann, U., Lüönd, F., and Mahrt, F.: *An introduction to clouds: From the microscale to climate*, Cambridge University Press, 2016.

Mason, B. J.: *Clouds, rain and rainmaking*, Cambridge University Press, 1975.

Pruppacher, H. R. and Klett, J. D.: *Microphysics of Clouds and Precipitation*, Springer Netherlands, Dordrecht, <https://doi.org/10.1007/978-0-306-48100-0>, 2010.

Rogers, R. R. and Yau, M. K.: A Short Course in Cloud Physics, Pergamon, Tarrytown, N. Y., 1989.

Watanabe, S., Hamilton, K., Sakazaki, T., and Nakano, M.: First Detection of the Pekeris Internal Global Atmospheric Resonance: Evidence from the 2022 Tonga Eruption and from Global Reanalysis Data, *Journal of the Atmospheric Sciences*, 79, 3027–3043, <https://doi.org/10.1175/JAS-D-22-0078.1>, 2022.

Woods, S., Lawson, R. P., Jensen, E., Bui, T. P., Thornberry, T., Rollins, A., Pfister, L., and Avery, M.: Microphysical Properties of Tropical Tropopause Layer Cirrus, *JGR Atmospheres*, 123, 6053–6069, <https://doi.org/10.1029/2017JD028068>, 2018.

Reviewer #3 (Remarks to the Author):

Review of "Transient ice ring observed during the 15 January 2022 eruption of Hunga volcano"

By Andrew T. Prata, Roy G. Grainger, Isabelle A. Taylor and Alyn Lambert

This study investigates a ring of cloud formed around the main volcanic plume after the underwater volcanic eruption from Hunga Tonga in 2021. Satellite retrievals show that the cloud consisted of very small ($\sim 2 \mu\text{m}$) ice crystals and lasted for approximately 2 hours. The authors hypothesize that this ice cloud was formed as a result of the Lamb wave that reduced the pressure and led to an adiabatic temperature decrease. The authors use meteorological data together with a model for ice crystal growth to show how an ice crystal cloud could form due to this temperature decrease.

The manuscript uses advanced processing of satellite data to both investigate the ice cloud and the Lamb wave propagation. The manuscript is in general well written and structured. However, in some parts of the text the language is a bit vague and not all features in the figures are explained. The topic of the paper is well within the scope of Communications Earth & Environment. The cloud of investigation is a unique feature that deserves investigation, but it has only been observed during this volcanic eruption. The authors motivate the importance of studying this event as a natural experiment to study ice crystal growth. However, I think there could have been more discussion of these processes in the manuscript. I also have some concerns regarding uncertainty estimates for the satellite retrievals and the ice crystal modelling. I would recommend this manuscript for publication if the author can address the following concerns and comments appropriately.

We thank Reviewer 3 (R3) for their detailed review. Addressing R3's comments has certainly led to improvements in the manuscript. In general we agree with the majority of R3's comments. We have added more discussion on ice cloud microphysics and provided further details on uncertainties in the satellite retrievals. We have provided direct responses to all (general and specific) below. It should be highlighted that in our revision we no longer directly attribute the Lamb wave to the generation of the temperature perturbation which led to the ice ring. Upon revision and in light of published research (in particular Watanabe et al., 2022, that we were not aware of at the original time of writing), we suggest that the Pekeris wave may have been the cause. However, attribution of the exact cause of the UTLS temperature wave (identifiable in the Himawari measurements) is beyond the scope of this study. Instead, we simply state that these are possibilities and our new ensemble modelling approach to the ice particle model attempts to cover the uncertainties in the magnitude of the UTLS temperature perturbation. All references mentioned in our responses are provided at the end of this document.

General comments:

All satellite measurements include some sort of uncertainty. I think the paper need more discussion with regards to the uncertainty of the size of the ice crystals. On page 8 you write “this process reveals a practical lower-limit for which the effective radius measurements can be retrieved using thermal infrared measurements.” Being close to the limits of what the retrieval can handle surely must mean that there are substantial uncertainties in the effective radius estimates.

The reviewer raises a good point here. This statement was poorly worded and we have removed it in our revision. What we meant by this was that these are the lowest ice particle sizes that have been reported (to our knowledge) using a passive thermal infrared retrieval technique (i.e. a ‘practical’ lower limit). But this is misleading as the reviewer rightly points out. For the semi-transparent optical depth that we see here ($\tau \sim 1-2$ at $11 \mu\text{m}$), small effective radii (r_{eff}) lead to a high brightness temperature differences (BTDs) that represent an excellent signal and lead to more stable retrievals when simultaneous solutions of r_{eff} and τ are retrieved. Uncertainties in optimal estimation retrieval schemes (such as ORAC) are also reduced for these optical depths (see Wang et al., 2016a, Fig. 9, for example).

Figure 4a shows results of retrievals of ice crystal radius over time. The error bars show the standard deviation of the data within the region of investigation. We do not however get any uncertainty estimated of the effective radius retrievals.

We have now added a new figure to the Supplementary Material which shows the mean effective radius uncertainty corresponding to the effective radii shown in Fig. 4a of the original manuscript (now Fig. 5 of the revised manuscript).

We have added a statement on the uncertainties involved for the particle size retrievals in the Method’s Sect. 3.1 as follows:

“Wang et al. (2016a) have shown that the uncertainty for thermal infrared retrieval algorithms reduces as the number of measurements increase and show that the uncertainty in the retrieved effective radius will be large for optically thick ($\tau \sim 10$) and optically thin ($\tau \sim 0.1$) ice clouds with the lowest uncertainties expected for semi-transparent ice clouds ($\tau \sim 1-5$).”

And we have added the following passage in the main text as follows:

“Figure 5 shows how the mean effective radius of the modelled ice particles compares with the satellite-retrieved mean effective radius within the ice-ring region. Note that the error bars on Fig. 5 represent the standard deviation of the satellite-retrieved effective radii and therefore are representative of the variation in effective radii within the ice-ring region. Supplementary Figure 2 provides the corresponding mean effective radius uncertainty determined from the satellite optimal estimation algorithm (see Methods Sect 3.1). The relative uncertainties for the satellite-retrieved effective radii range from 14-36 %. Additionally, as the optical depth increases, the uncertainty on the effective radius decreases and vice-versa, which is expected for thermal infrared optimal estimation retrieval algorithms applied to semi-transparent ice clouds (Wang et al., 2016a,b)”

If the retrievals have an uncertainty of even 10 % then that will cover a quite large range of the ppmv estimates from the model shown in Figure 3a. Thus, knowing the uncertainty range is crucial for estimating model performance.

Four points should be made here:

1) To better quantify uncertainties in ambient conditions, we now conduct ensemble modelling experiments which cover a range of environmental conditions to better quantify the uncertainty in the modelled ice particle size (note we now model a size distribution rather than a single particle).

2) We are not attempting to evaluate the ice particle model with the satellite retrievals (i.e. 'assess model performance'). Rather, we use the ice particle model in an attempt to explain the ice cloud microphysics and assess where or not a UTLS temperature perturbation identified in the satellite observations can explain the small ice particle sizes retrieved from the ORAC algorithm.

3) We now provide the effective radius uncertainties as a supplementary figure. We find relative uncertainties from 14-36%, which is quite low for this type of satellite retrieval. For comparison, Wang et al. (2016b) find ice effective radius uncertainties ranging from ~10-50% (see their Fig. 5, row 3).

4) In our supplementary figure, we also provide the optical depth retrieval to show that as the optical depth increases the uncertainty on the effective radius generally decreases and vice-versa (consistent with the Wang et al. (2016a, b) findings).

There are quite large error bars (variation) for the satellite measurements and the measurements do not follow quite the same pattern as the red line. Is it really possible to deduce which ppmv is the best fit to such a high accuracy as you state in the manuscript?

No, we do not believe that we can deduce a water vapour concentration to such high precision and did not intend to imply this in our original manuscript. We simply reported the modelled value that resulted in the best fit. We have now revised our processing of the satellite data to produce an improved ice ring effective radius time series. Previously we had just set a threshold BTD of 15 K and used the annulus region between 250-300 km to identify the ice-ring cloud. We now apply 4 criteria (see revised Methods section 3.1), which has resulted in less masked pixels and more robust estimates of mean effective radii in the ice ring. Further, we have completely redesigned the modelling experiments where we now model a distribution of ice particles, and vary the environmental conditions (different ambient temperature, pressure, specific humidity etc), the UTLS temperature perturbation and ice mass accommodation coefficient.

There is quite some discussion of the vapor concentrations needed for the formation of the ice crystals. An important aspect of the ice crystal formation that is not discussed at all is the initial size of the ice crystals before they start growing. The manuscript simply states that the authors assume an initial particle radius of 0.01 μm . Can you justify this starting size?

Upon revision and further analysis of the satellite data, we now argue that the atmosphere was supersaturated with respect to ice before the 15 January eruption. This argument, which was only briefly suggested in the original manuscript, addresses the question of the source of ambient water vapour and also the source of ice nucleating particles. In our revision, we suggest these could be ash particles and justify the choice of the size distribution of particles based on aircraft measurements presented in Schumann et al. (2011) and Johnson et al (2012) and lidar observations and model simulations in Rolf et al. (2012).

What type of particles of this size would be present in the upper troposphere?

We argue that the particles may have been ash particles because they are known to act as efficient ice nuclei (Durant et al., 2008; Rolf et al., 2012), although we cannot rule out different particles such as sea salt.

The size of the initial seeds will have a large impact on the growth of the crystals and the fit of the crystal sizes to the observations. I think the choice of this size needs a more thorough motivation.

We now model a size distribution of ash particles and justify their sizes based on aircraft measurement of airborne ash. The shape of the size distribution has been observed to be lognormal and number concentrations reported are 0.1 cm^{-3} (Schumann et al., 2011; Rolf et al., 2012). From Rolf et al. (2012): "Therefore a suitable assumption for the simulation of the volcanic-ash-induced cirrus is a high IN concentration of 0.1 cm^{-3} ".

It would also be interesting to perform a sensitivity analysis of the initial particle size and its importance for the growth over time. Using larger particles would most likely require less moisture to reproduce the growth in the observations.

In our revision we now model a size distribution of particles in a unit volume of air. The simulations are conducted by simulating a single ice particle growth for each particle in the distribution which covers a range of initial dry radii sizes. We also account for the change in saturation ratio as the ice particles grow and compete for available background water vapour (see revised Methods Sect. 3.4). We also compute the effective radius from the distribution of particles to produce a better "like for like" comparison with the satellite effective radius retrievals.

I think the manuscript would also need a discussion of what particles that could be present to initiate heterogeneous ice nucleation and if enough particles could be present to form such a large ice cloud. A discussion if homogeneous ice formation could be responsible for such a cloud would also be suitable in this manuscript.

In our revision we suggest that ash particles from the 13-14 January activity were the likely source of particles for the ice ring that appeared on 15 January. To further back this argument, we have also included a new figure that shows prior to the 15 January eruption a large ice cloud was present over the region over Hunga. The BTDF figures show that the atmospheric waves generated by the eruption caused a sublimation of these ice particles and then re-nucleation of them which formed the ice ring of interest here. This additional evidence addresses both R3's comment regarding the possibility of homogenous ice nucleation and the presence of enhanced water vapour (see discussion below). As ice particles were present and then sublimated, there must have been INPs ready to nucleate as the cold part of the UTLS temperature wave propagated through the atmosphere.

The authors model for ice crystal growth requires more vapor present at 100 hPa than the ERA data show at this level. The authors speculate that the UTLS water vapor was enhanced by the 13 January explosion (2 days prior to the formation of the cloud). However, atmospheric transport in the UTLS in this region is impacted by the subtropical jet and other high-level winds. Water vapor and particles emitted in the UTLS will be transported eastward and would most likely not be present in the region over Hunga Tonga 48 hours later. Have you taken this into consideration?

Yes we have considered this - R3's assumption here is that the ice particles were generated and then there was no more activity after 13 January. In fact, in our Supplementary Movie 1, we show that ice was being produced almost continuously into 14 January and in the lead up to the 15 January event. Supplementary Movie 1, along with our new Figure 3 provide compelling evidence that ice was present over the volcano before the 15 January eruption and was a result of the volcanic activity from 13-14 January. We also take this situation into account in our new ensemble modelling approach.

Specific comments:

Page 1 line 036: "Record breaking heights have been reported..". Heights of what? This is vague. Heights of the volcanic plume. We have added this clarification.

Page 2 line 083: "only modest amounts of SO₂ were measured" Was this in the stratosphere or in the atmosphere in general?

In the atmosphere in general. We have rephrased the whole sentence to avoid any confusion and have added updated references and reference to Sellitto et al. (2024) who provide further estimates of the SO₂ from the 15 January eruption:

"Significant amounts of H₂O (~130–150 Tg) were injected into the stratosphere according to microwave limb sounder retrievals (Millan et al., 2022; Khaykin et al., 2022; Xu et al., 2022; Nedoluha et al., 2024). Initial estimates of SO₂ mass determined from ultraviolet vertical column density retrievals were modest [~0.4–0.5 Tg; Carn et al., 2022], but more recently hyperspectral thermal infrared measurements indicate a total mass of ~1 Tg SO₂ for the whole event (Sellitto et al., 2024)".

Page 3 line 101-103. It would have been very nice to see the evolution of the ice ring in satellite data images to give the reader a better view of the development. Maybe such a figure could be included in a supplementary.

We have included a new animation (Supplementary Movie 2) which covers the time from 04:07 to 07:07 UTC on 15 January 2022.

Page 3 lines 111-114. You claim that the very large difference in BTD between the 8.6 μm and 11.2 μm indicates small crystal sizes. It would be nice to have an explanation of why.

The reason is comes from Mie theory, which is used in radiative transfer calculations at thermal infrared wavelengths and particle sizes of the order of microns. We have revised the statement and provided relevant references which provide further details and modelling to support this statement: “According to well-established radiative transfer theory, the exceptional signal (at times > 50 K) in the brightness temperature differences (BTDs) between the 8.6 and 11.2 μm channels (Fig. 1b) indicates that the ice ring was semi-transparent, composed of small ice particles and there was a strong thermal contrast between the surface and the temperature at the height of the cloud (Rose et al., 1995; Heidinger & Pavolonis, 2009; Wang et al., 2011; Prata et al., 2020).”

To alleviate any further concerns of the reviewer, consider the Fig. 4c from Wang et al. (2011):

FIG. 4. Simulated TOA BTDs with respect to 11- or 12- μm BTs. The ice cloud is located between 8 and 8.5 km in a generalized tropical atmosphere.

Here one can see that for a given optical depth, the BTD increases with decreasing ice effective particle diameter.

Page 3 lines 120 - 121. I agree that ice crystals of 2 μm are very small. However, calling them extremely small and then in the next part of the sentence write that tropical tropopause cirrus clouds are 2.5 to 25 μm seems a bit contradictory. There is not a very large difference between 2 and 2.5 μm . We agree and have completely revised this section. See revised Sect. 2.1 for the full and more detailed discussion on ice cloud microphysics and typical particle sizes.

Page 3 lines 133-136: It would have been very nice to show these ice mass fluctuations in a figure. We have now provided a new figure (Supplementary Figure 1) showing the ice mass time series. We have also removed the statement “Additionally, a wave structure is observed in the ice mass time-series” to avoid overinterpreting the data. We now simply state that the ice mass time series is shown in Supplementary Figure 1.

Page 5 lines 189-190. You write that the mean temperature reduction is 6 K and that the maximum reduction is 8 K. However, the temperature scale in the figure is between -5 and +5 K. Thus these results cannot be seen in the figure. Please change the color scale.
Done. Colour scale has been changed to range from -9 K to +9 K.

Page 5 lines 190 to 192: This sentence is confusing. We have no prior information about a ozone weighting function or what is used for. Please consider rewriting this sentence. We have provided a further explanation of what a weighting function is and how it relates to the ozone concentration in the atmosphere. We have also revised Fig. 2b to indicate the full-width half maximum range. The revised text is as follows:
“The weighting function is defined as the derivative of the transmittance profile with respect to the vertical coordinate (e.g. height or pressure) for a given wavelength or wavelength interval for a broadband infrared channel (Rodgers, 2000). Weighting functions therefore represent the relative contributions of each atmospheric level to the measured top-of-atmosphere radiance in a particular channel. The 9.6 μm channel is sensitive to ozone absorption and so its weighting function is tied to the vertical distribution of ozone in the atmosphere (Prabhakara et al. 1970). We estimate that the temperature perturbations shown in Fig. 2a would correspond broadly to pressure heights from 6-52 hPa (21-35 km) based on the 9.6 μm channel weighting function [...]”

Fig 2a line 215: It is confusing that you have rings at different distance from the volcano in this figure compared to Figure 1 and 3. It would be easier for the reader if you kept the solid circle marking the same area in this figure as in Figure 1 and 3 and show the other features related to the Lamb wave with a differently formatted ring.

Based on this comment and related comments from R1 and R2 we have removed Fig. 3 from the revised manuscript as it is no longer necessary. We have revised Fig. 2 of the original manuscript as follows: We have removed the two dashed-line circles on the original Fig. 2a and now plot two circles corresponding to the predicted location of the Lamb wave and Pekeris wave. We have also added the 300 km radius from Fig.1 onto revised Fig. 2a to improve clarity as suggested by R3.

Fig 2 line 220: What is RTTOV? This acronym is not explained or used in any other part of the paper. RTTOV is a radiative transfer model. We have now spelt it out and provided a reference.

Page 6 line 254-256: This sentence is confusing. Please consider rewriting it.

We have removed it in our revision as this section of the paper has been completely restructured.

Page 7 line 309: You use the word model in quite a lot of contexts in the manuscript. It would be good to specify here which of the models you are referring to.

It's the 'ice particle model' here. We have added this to clarify.

Page 7 line 313: The ERA data showed 4-5 ppmv but here you instead use 5-6 ppmv without explaining why.

In our original manuscript we suggested this enhancement was due to the sublimation of ice particles generated by the 13 January eruption (later on line 330 of the original manuscript). In our revision, we explain this in more detail and provide a new figure (Fig. 3) and Supplementary Movie 1 to justify our argument that the atmosphere was supersaturated with respect to ice prior to the 15 January eruption.

Page 7 line 317: "can mostly explain" What do you mean by mostly here? Are there other factors that can explain this? The expression is vague and a bit confusing.

We agree and have removed this discussion in our revision. We have completely revised this section of the paper to describe the results of our new ensemble modelling approach. See revised Sect. 2.5.

Fig 4b line 398: What are the grey dashed lines shown in this figure? Where does the data for the red and blue bars come from?

In Fig. 4 of the original manuscript, the grey dashed lines indicated the frost point temperature (at 100 hPa) for different water vapour concentrations. They were used to show the temperature that the ambient temperature would be required to reach for different ambient water vapour concentrations. The data for the red and blue bars comes from the satellite observations (Himawari). We have revised and simplified Fig. 4 in our revision. We have added further details on how this figure was constructed in the revised figure caption and in revised Sect. 3.2.

Page 13 line 583: Do you model the growth of one ice crystal or many. In case you model many, in what volume of air do you then model them?

We originally modelled one ice particle (as stated on page 6, line 267 of the original manuscript). However, in our revised manuscript, we have extended our model to consider a size distribution of particles in a unit volume of air (see revised Sect. 3.4).

References

Carn, S. A., Krotkov, N. A., Fisher, B. L., and Li, C.: Out of the blue: Volcanic SO₂ emissions during the 2021–2022 eruptions of Hunga Tonga—Hunga Ha’apai (Tonga), *Front. Earth Sci.*, 10, 976962, <https://doi.org/10.3389/feart.2022.976962>, 2022.

Durant, A. J., Shaw, R. A., Rose, W. I., Mi, Y., and Ernst, G. G. J.: Ice nucleation and overseeding of ice in volcanic clouds, *Journal of Geophysical Research*, 113, <https://doi.org/10.1029/2007JD009064>, 2008.

Heidinger, A. K. and Pavolonis, M. J.: Gazing at Cirrus Clouds for 25 Years through a Split Window. Part I: Methodology, *Journal of Applied Meteorology and Climatology*, 48, 1100–1116, <https://doi.org/10.1175/2008JAMC1882.1>, 2009.

Johnson, B., Turnbull, K., Brown, P., Burgess, R., Dorsey, J., Baran, A. J., Webster, H., Haywood, J., Cotton, R., Ulanowski, Z., Hesse, E., Woolley, A., and Rosenberg, P.: In situ observations of volcanic ash clouds from the FAAM aircraft during the eruption of Eyjafjallajökull in 2010, *J. Geophys. Res.*, 117, 2011JD016760, <https://doi.org/10.1029/2011JD016760>, 2012.

Khaykin, S., Podglajen, A., Ploeger, F., Grooß, J.-U., Tence, F., Bekki, S., Khlopenkov, K., Bedka, K., Rieger, L., Baron, A., Godin-Beekmann, S., Legras, B., Sellitto, P., Sakai, T., Barnes, J., Uchino, O., Morino, I., Nagai, T., Wing, R., Baumgarten, G., Gerding, M., Duflot, V., Payen, G., Jumelet, J., Querel, R., Liley, B., Bourassa, A., Clouser, B., Feofilov, A., Hauchecorne, A., and Ravetta, F.: Global perturbation of stratospheric water and aerosol burden by Hunga eruption, *Commun Earth Environ*, 3, 316, <https://doi.org/10.1038/s43247-022-00652-x>, 2022.

Millán, L., Santee, M. L., Lambert, A., Livesey, N. J., Werner, F., Schwartz, M. J., Pumphrey, H. C., Manney, G. L., Wang, Y., Su, H., Wu, L., Read, W. G., and Froidevaux, L.: The Hunga Tonga-Hunga Ha’apai Hydration of the Stratosphere, *Geophysical Research Letters*, 49, <https://doi.org/10.1029/2022GL099381>, 2022.

Nedoluha, G. E., Gomez, R. M., Boyd, I., Neal, H., Allen, D. R., and Lambert, A.: The Spread of the Hunga Tonga H₂O Plume in the Middle Atmosphere Over the First Two Years Since Eruption, *JGR Atmospheres*, 129, e2024JD040907, <https://doi.org/10.1029/2024JD040907>, 2024.

Prata, A. T., Folch, A., Prata, A. J., Biondi, R., Brenot, H., Cimarelli, C., Corradini, S., Lapierre, J., and Costa, A.: Anak Krakatau triggers volcanic freezer in the upper troposphere, *Scientific Reports*, 10, <https://doi.org/10.1038/s41598-020-60465-w>, 2020.

Rodgers, C. D.: *Inverse methods for atmospheric sounding : theory and practice*, World Scientific, Singapore, 2000.

Rolf, C., Krämer, M., Schiller, C., Hildebrandt, M., and Riese, M.: Lidar observation and model simulation of a volcanic-ash-induced cirrus cloud during the Eyjafjallajökull eruption, *Atmos. Chem. Phys.*, 12, 10281–10294, <https://doi.org/10.5194/acp-12-10281-2012>, 2012.

Rose, W. I., Delene, D. J., Schneider, D. J., Bluth, G. J. S., Krueger, A. J., Sprod, I., McKee, C., Davies, H. L., and Ernst, G. G. J.: Ice in the 1994 Rabaul eruption cloud: implications for volcano hazard and atmospheric effects, *Nature*, 375, 477–479, <https://doi.org/10.1038/375477a0>, 1995.

Schumann, U., Weinzierl, B., Reitebuch, O., Schlager, H., Minikin, A., Forster, C., Baumann, R., Sailer, T., Graf, K., Mannstein, H., Voigt, C., Rahm, S., Simmet, R., Scheibe, M., Lichtenstern, M., Stock, P., Rüba, H., Schäuble, D., Tafferner, A., Rautenhaus, M., Gerz, T., Ziereis, H., Krautstrunk, M., Mallaun, C., Gayet, J.-F., Lieke, K., Kandler, K., Ebert, M., Weinbruch, S., Stohl, A., Gasteiger, J., Groß, S., Freudenthaler, V., Wiegner, M., Ansmann, A., Tesche, M., Olafsson, H., and Sturm, K.: Airborne observations of the Eyjafjalla volcano ash cloud over Europe during air space closure in April and May 2010, *Atmos. Chem. Phys.*, 11, 2245–2279, <https://doi.org/10.5194/acp-11-2245-2011>, 2011.

Sellitto, P., Siddans, R., Belhadji, R., Carboni, E., Legras, B., Podglajen, A., Duchamp, C., and Kerridge, B.: Observing the SO₂ and Sulfate Aerosol Plumes From the 2022 Hunga Eruption With the Infrared Atmospheric Sounding Interferometer (IASI), *Geophysical Research Letters*, 51, e2023GL105565, <https://doi.org/10.1029/2023GL105565>, 2024.

Wang, C., Yang, P., Baum, B. A., Platnick, S., Heidinger, A. K., Hu, Y., and Holz, R. E.: Retrieval of Ice Cloud Optical Thickness and Effective Particle Size Using a Fast Infrared Radiative Transfer Model, *Journal of Applied Meteorology and Climatology*, 50, 2283–2297, <https://doi.org/10.1175/JAMC-D-11-067.1>, 2011.

Wang, C., Platnick, S., Zhang, Z., Meyer, K., and Yang, P.: Retrieval of ice cloud properties using an optimal estimation algorithm and MODIS infrared observations: 1. Forward model, error analysis, and information content: IR-Based Ice Cloud Retrieval Algorithm, *Journal of Geophysical Research: Atmospheres*, 121, 5809–5826, <https://doi.org/10.1002/2015JD024526>, 2016a.

Wang, C., Platnick, S., Zhang, Z., Meyer, K., Wind, G., and Yang, P.: Retrieval of ice cloud properties using an optimal estimation algorithm and MODIS infrared observations: 2. Retrieval evaluation: Comparison of Ice Cloud Retrievals, *Journal of Geophysical Research: Atmospheres*, 121, 5827–5845, <https://doi.org/10.1002/2015JD024528>, 2016b.

Watanabe, S., Hamilton, K., Sakazaki, T., and Nakano, M.: First Detection of the Pekeris Internal Global Atmospheric Resonance: Evidence from the 2022 Tonga Eruption and from Global Reanalysis Data, *Journal of the Atmospheric Sciences*, 79, 3027–3043, <https://doi.org/10.1175/JAS-D-22-0078.1>, 2022.

Xu, J., Li, D., Bai, Z., Tao, M., and Bian, J.: Large Amounts of Water Vapor Were Injected into the Stratosphere by the Hunga Tonga–Hunga Ha’apai Volcano Eruption, *Atmosphere*, 13, 912, <https://doi.org/10.3390/atmos13060912>, 2022.

Response to:

Second review of

Transient ice ring observed during the 15 January 2022 eruption of Hunga volcano

Andrew T. Prata¹, Roy G. Grainger, Isabelle A. Taylor and Alyn Lambert

Previous reviewer comments are in **black**, previous author responses in **blue**, second reviewer comments are in **green** and final responses are in **red**.

General:

 The manuscript has improved significantly. The authors invested considerable effort into the study, and their work has paid off. The paper is now almost ready for publication. I only have a few comments/questions -both on previously raised issues and on some new points- which are listed below.

Specific comments:

 1) line 120: 'The mean ice particle sizes were extremely small (Fig 1d) when compared to tropical tropopause cirrus clouds [2.5–25 μm radius; 20].'

2 μm ; ~The size of ice crystals depend on the stage of their development. All in-situ formed ice particles appear at very small sizes, because they form either heterogeneously on INPs or homogeneously on supercooled solution particles and thus have initially about the size of the particles, which is $< 1 \mu\text{m}$. In the course of the evolution of the ice cloud, they grow as long as the environment is supersaturated. The growth depends on the number of ice crystals, since the ice crystals compete for the available water vapor.

At the cold temperatures and low water vapor mixing ratios prevailing in the icing, the ice particles cannot grow to mean sizes substantially larger than $2 \mu\text{m}$, especially since the time for growth is short (see your line line 120: 'the whole process (from formation to dissipation) took 2 h to complete'). You show this in Figure 4 (the cooling / growth phase is only about 10min). So the reason that in most cases the ice particles grow larger under natural TTL conditions is that the growth periods are longer, or that the ice crystal concentration is very small. A typical mean ice crystal size in the TTL is $\sim 8 \mu\text{m}$ (Krämer et al., 2020, ACP).

- I recommend to discuss in the paper that a shorter growth periods and/or a larger ice crystal concentration is likely the reason for the smaller ice particles observed in the ice ring in comparison to natural TTL ice clouds.

- Also, I recommend not saying that the ice crystals are extremely small, but that such small ice crystals are unlikely to occur in the atmosphere under natural conditions.

Thank you for these comments and suggestions. We appreciate these insights and have incorporated them into our revised manuscript. We couldn't find a statement or discussion in Krämer et al. (2020) stating that "typical mean ice crystal size in the TTL is 8 μm ". Looking at their figures (see Figure S2 for R_{ice}), it looks like the median at 15-18 km is colour-coded according to the 6-18 μm radius range.

The mean ice crystal size of $\sim 8 \mu\text{m}$ can be seen in Krämer et al. (2020), Figure 8, panel f, at the lowest temperatures ($<190\text{K}$).

Thank you. We have added this information to the revised manuscript.

We have revised the statement to make mention of both factors discussed (namely, ice crystal concentration and growth duration) and have deleted any mention of "extremely small" ice particles. The relevant revised text is: "The mean ice particle effective radii within the ice ring at 05:17~UTC was $\sim 2 \mu\text{m}$ (Fig. 1d). Tropical tropopause cirrus cloud sizes have been reported in the range from 2.5-25 μm radius (Woods et al., 2018); however, the size observed depends upon the stage of cloud development. The Hunga ice ring particle sizes would fall into the smallest, median R_{ice} size range (1-6 μm) reported by Krämer et al. (2016, 2020) who compiled a cirrus ice particle size climatology from in situ aircraft data. Krämer et al. (2020) show that the median ice particle sizes for cirrus clouds at similar altitudes ($\sim 15\text{-}18 \text{ km}$) lie in the range from 6-18 μm ." We have also added a footnote to explain how R_{ice} is defined.

2) line 123 ff: '.. ; however, the ice particles studied here are of an order of magnitude smaller, indicating a new ice nucleation pathway not previously documented for volcanic clouds.'

I think this statement is somewhat exaggerated - the observations can certainly be explained by conventional ice formation mechanisms - see the previous comment (on line 120).

In my opinion, the difference to previous volcano-induced ice clouds lies in the short period of the Lamb wave, and possibly in the number of ice crystals (see also previous comment). Is there any information about this? That would be helpful, and it would also be advisable to estimate the time scales for ice growth during previous volcanic eruptions.

Nevertheless, I find the observation and description of this phenomenon to be important.

We were unable to find published estimates of the timescales (e.g. in seconds or minutes) for ice particle growth in volcano-induced ice clouds. Durant et al. (2008), which we cite, provides a comprehensive overview of volcano-induced ice clouds. We agree with the reviewer that a possible reason for the small ice particle sizes is due to the short duration of ice

supersaturation triggered by the atmospheric waves caused by the eruption. In response to this reviewer comment we have revised and added the following text:

“Ice particle sizes of $\sim 20 \mu\text{m}$ associated with volcanic eruptions have been observed before (Durant et al., 2008; Prata et al., 2020) and the particle sizes observed have been attributed to overseeding (i.e. high number concentrations of ice nucleating particles; Durant et al., 2008); however, the ice particles studied here are of an order of magnitude smaller, indicating an earlier stage of ice particle growth that has not previously been documented for ice clouds associated with volcanic eruptions. Such small ice particles are unlikely to be observed under natural conditions. In theory, the smallest ice particles will be observed at the very beginning of the ice particle growth phase during cloud development. Once nucleated, ice particles will initially grow via vapour deposition. Particle growth via vapour deposition is principally driven by the level of supersaturation, initial (dry) particle size distribution and, **in case of heterogeneous ice nucleation***, the ice nucleating particle (INP) number concentration (Mason, 1975; Rogers and Yau, 1989, Pruppacher and Klett, 2010; Lohmann et al., 2016). The longer an environment remains supersaturated with respect to ice, the larger an ice particle will be expected to grow. Thus if the environment is subject to only a short period of supersaturation then small ice particles may be expected. For distributions of ice particles, low number concentrations will lead to larger particle sizes (and vice versa) as individual ice particles compete for the available background ambient water vapour.”

* this is important to mention because in homogeneous ice nucleation –the other pathway of ice formation- the number of ice particles does not depend on the number of homogeneously freezing aerosol particles. I would also order the sentences by importance of the parameter: ... driven by the level of supersaturation, and, in case of heterogeneous ice nucleation, the ice nucleating particle (INP) number concentration, and the initial (dry) particle size distribution.’

Done.

4) line 229ff: 'It is common in volcanic eruptions for water vapour in the atmosphere to be entrained into the plume, condense to liquid water and freeze in updrafts during transport to plume-top.'

Freezing of liquid drops is an additional way to form ice clouds in comparison to the two ways explained above (see e.g. Luebke et al., 2016, ACP). It is mostly triggered by INPs (thus heterogeneous), but liquid cloud drops would for sure be larger than $2 \mu\text{m}$! Plus, once frozen, the ice particles quickly grow to much larger sizes. In case of high updrafts, liquid drops can survive until they reach the temperature of $\sim -40\text{C}$ and then freeze homogeneously - but even then the sizes would be around $10\text{-}20 \mu\text{m}$. So I don't think that this pathway is possible here ...

Maybe this was the way the ice particles have formed in earlier volcanic eruptions ? (see also comment 3)

- I recommend to consider and explain in more detail the different ways of ice formation, as mentioned above.

To address both comments (3) and (4) we have revised Section 2.3 by:

1. Mentioning both pathways of ice formation (heterogeneous INPs and supercooled solution aerosol particles).
2. Mentioning the Rolf et al. (2012) study which shows how volcanic particles are efficient INPs.
3. Adding further discussion on ice cloud (cirrus) formation drawing on studies from Luebke et al. (2016), Krämer et al. (2016, 2020), Lawson et al. (2019) and Karcher et al. (2022). Our revised text is as follows:

“In order for ice clouds to form in the atmosphere there needs to be a source of liquid water or water vapour that is brought to supersaturation with respect to ice. For cirrus, recent work suggests that these clouds can be grouped into two broad categories: in situ origin cirrus and liquid origin cirrus (Luebke et al., 2016; Krämer et al., 2016, 2020). In situ origin cirrus clouds may form heterogeneously on ice nucleating particles (INPs) or may form homogeneously on supercooled solution aerosols (Kärcher et al., 2022 – there are much earlier fundamental references, e.g. Pruppacher and Klett, 1997). Liquid origin cirrus clouds typically contain larger ice particles than in situ origin cirrus as they form from liquid water drops which rapidly grow once frozen (Luebke et al., 2016; Lawson et al. 2019). Even in the case of strong updrafts, where liquid water droplets can survive until they reach $-38\text{ }^{\circ}\text{C}$ and freeze homogeneously, the smallest liquid origin cirrus ice particle sizes are typically $\sim 10\text{-}20\text{ }\mu\text{m}$ radius (Krämer et al., 2016).

Thanks. We have added reference to Pruppacher and Klett (1997).

Volcanic ash particles are known to be efficient INPs (Durant et al. 2008; Rolf et al. 2012) and it is common in volcanic eruptions for water vapour in the atmosphere to be entrained into the plume, condense to liquid water and freeze in updrafts during transport to plume-top (Woods et al. 1993; Tupper et al. 2009). Magmatic water, expelled during an eruption, is also a source of water for ice nucleation, meaning that volcanogenic ice can form in relatively dry atmospheres (Rose et al. 2004). A further source of water vapour can be added to the atmosphere via phreatomagmatic interactions (Prata et al. 2020). It is likely that all of the aforementioned mechanisms were at play during the Hunga eruption, which raises the question: Which ice nucleation pathway was responsible for the formation of the ice ring?”

New comment

1) line 316 ff of the revised version : ,... we explored the hypothesis that the temperature perturbation generated by atmospheric waves in this altitude region triggered rapid, large-scale heterogeneous deposition nucleation in the UTLS.’

How sure are you that it was pure heterogeneous ice nucleation? It is not unlikely that ice formation initially begins heterogeneously, but then homogeneous freezing of supercooled solution aerosols sets in (see Rolf et al., 2012, their Figure 7). Perhaps both possibilities should be described.

We have listed this ice nucleation pathway based on your previous reviewer comment: “In situ origin cirrus clouds may form due to heterogeneous ice nucleation or via homogeneous freezing of supercooled solution aerosols”.

However, as stated, the hypothesis we have put forward is that ice nucleation was triggered heterogeneously due to the presence of pre-existing INPs (likely ash particles) in the atmosphere. This hypothesis is supported by:

1. Evidence of ash deposition on the islands of Hunga Tonga-Hunga Ha’apai (Colombier et al., 2023) following the January eruptions.
2. Evidence of ice particles in the atmosphere before the 15 January eruption.
3. Previous studies demonstrating that volcanic ash particles can act as efficient ice nuclei.

Based on this hypothesis, we are able to model what is indicated by the satellite retrievals. In our study, we do not explicitly rule out homogenous aerosol freezing. This hypothesis could be explored further in a different study beyond the scope of the present study.

Reference:

Colombier, M. et al. Atmosphere injection of sea salts during large explosive submarine volcanic eruptions. *Scientific Reports* 13 (1), 14435 (2023). URL <https://www.nature.com/articles/s41598-023-41639-8>.
<https://doi.org/10.1038/s41598-023-41639-8> .

New comment

2) line 427f of the revised version: ‘... four levels of ice supersaturation (0.1, 1, 5 and 10 %)...’ I wonder if the supersaturation is kept constant during the simulations or if it adjusts? That would be good to mention in the text.

These are the initial supersaturations that we tested in the ensemble modelling. The supersaturations then vary in time according to Eqs. (16) to (23) in the Methods section. We have added this detail to the main manuscript and added the word “initial” when referring to the supersaturation model runs to improve clarity.

New comment

3) line 500f of the revised version: the presence of pre-existing ice clouds (and INPs) were necessary to explain the supersaturated conditions required to form the ice ring.

Is this an assumption or a fact? Wouldn't the temperature drop caused by the wave have increased the supersaturation sufficiently to trigger ice formation even without previously evaporated clouds? According to Fig. 4, the temperature dropped by 5 degrees, which corresponds to an increase in the relative humidity above ice of approximately 50%. It would be great if this could be discussed more clearly.

As stated in the manuscript, based on our modelling and the satellite observations, we find that the environment must have been supersaturated to grow ice particles consistent with the size and timescale indicated by the satellite retrievals. Consider the case of the initial 0.1 % supersaturation (Fig. 5a of the manuscript), even with 0.1 % supersaturation, the particles cannot grow large enough to be consistent with the satellite measurements. The reviewer has misinterpreted Fig 4. **We do not find nor model a temperature drop of 5 K at the height of the ice ring (16-18 km).** Fig. 4 shows the temperature change in the lower-middle-stratosphere (LMS; 21-35 km). **This is not the height region where the ice ring formed.** The purpose of Fig. 4 is to show how the temperature evolved with time (i.e. justifying the harmonic oscillator model). We were careful to explicitly state these subtleties in the manuscript:

“The introduction of an abrupt pressure pulse at the surface is theorised to cause a change in temperature throughout the entire atmospheric profile and the magnitude of this temperature change increases exponentially with height [42, 48, 50, 54]. **Therefore, it is expected that the magnitude of the temperature change due to the Hunga eruption would be lower at altitudes where the ice ring formed (75–115 hPa; 16–18 km).** Although the precise amplitude of the temperature perturbation at these altitudes is difficult to determine, Fig. 4 shows both the temperature perturbation time series (red and blue bars) and the temperature change with time relative to the background temperature (black data points). The term ‘back-ground temperature’ is used to refer to the temperature of the LMS in the ice-ring region at 04:47 UTC (i.e. before the large cold-ring anomaly, shown in Fig. 2a, is detected). What is immediately apparent is that the temperature times series can be well described by the equation for a damped harmonic oscillator (black solid line in Fig. 4).”

We also noticed on line 694, we state:

“Since we do not know the precise amplitude of the temperature change in the UTLS (i.e. the minimum of $T_{LMS}(t_i)$), ...”. This is unnecessarily confusing - what we meant was “the minimum of the temperature time-series in the UTLS is unknown”.

To improve clarity, we simply removed the statement in parentheses:

“Since we do not know the precise amplitude of the temperature change in the UTLS, ...”

Review of Transient ice ring observed during the 15 January 2022 eruption of Hunga volcano

The manuscript describes satellite observations of an expanding ring of extremely fine ice crystals which formed in the upper troposphere/lower stratosphere in response to the 15 January 2022 Hunga eruption, Tonga. This is a uniquely observed atmospheric phenomenon. By combining radiometric satellite observations with previously published observations of the atmospheric waves generated by the eruption, the authors hypothesise that the ice ring formed due to the passage of the generated Lamb wave. Finally, the authors support this hypothesis by developing a model of ice crystal growth in response to the Lamb wave, showing agreement between the predicted ice crystal sizes and their timescale of formation.

As has been reported in many published articles, the 15 January 2022 Hunga eruption produced many phenomena unique within the period of modern observations. The ice ring described in this manuscript has not been previously described before. That the authors can support this observation with a quantitative model of formation is a strength of the paper and makes this a well-rounded study. Reading the paper, it became apparent that the study is more about atmospheric processes triggered by the eruption, rather than the eruption itself. Therefore, I think that the manuscript is of more specific interest to atmospheric scientists rather than volcanologists. However, given the uniqueness of the eruption, I think the paper will be of general interest to a wider audience of Earth and atmospheric scientists.

I have no serious concerns about the paper. My biggest issue is that I think there are a couple of areas where the observations and methods can be made a little clearer to the readers. I detail these below but, as an example, it is not visually clear from Fig 1. that the ice ring is a) a ring at all rather than a disc and b) propagating outwards (see major comment 3). Both of these could be addressed by showing a timeseries of satellite images. A second issue is that I think the authors need to be careful with how they compare their modelled results to the observations (see major comments 5 and 6).

To conclude, I think this is a good paper which does an excellent job of interrogating a novel observation, presenting a quantitative model to explain it. There are some minor areas where the manuscript can be improved, as summarised in the above paragraph and detailed below. In the following, I provide some detailed comments, separated into those on the scientific content, and those on the manuscript presentation and style. I want to note that I only provide comments on the presentation and style to help the authors as they revise the manuscript. I have also provided a short list of references at the end of this review of papers which I refer to in the comments below. Whilst I think all of these may be useful for the authors as they revise the manuscript, I want to make clear I am not asking for them to be added as references to the paper.

Major comments

1. Lines 60-61. This may be ignorance on my part, but why do umbrella clouds and overshooting tops indicate that a volcanic plume reached the tropopause? My understanding is that these will form if 1) a plume is able to reach a neutral buoyancy level and 2) the plume is strong, i.e., the ambient wind is insufficiently strong to significantly impact the plume rise. Is it not possible for both of these conditions to be true for a plume which does not reach the tropopause?
2. Lines 74-75. In fact, eruptive activity commenced on 19 December 2021, and not 13 January 2022 (Gupta et al., 2022).
3. Lines 88-91 and Figure 1. After reading this description of the ice ring, it took me a little while to work out that the ice ring is actually the grey, diffuse cloud beneath the uppermost umbrella in Figure 1a. I think this is because, to me, it does not look very ring-like. Instead, to me, it looks like a disc-shaped umbrella cloud, with the inner portion visually obscured by the higher umbrella. In fact, from the single image shown in Figure 1a alone, it is impossible for the reader to identify this cloud as a ring with an inner and outer diameter, rather than a disc which is continuous to the vent location. To help the reader here, it may help to show a timeseries of true colour images (and maybe the other derived images as well), from which the reader will be able to identify the ring shape and see the expansion.
4. Line 196 and Figure 2a. You refer to a Lamb wave initiated at 04:28 UTC, as described by Wright et al. (2022). Inspection of Wright et al.'s Extended Data Fig 1e shows that this origin time was determined by recording the time of peak pressure disturbance at ground level stations, and performing a linear regression back to Hunga volcano. Jarvis et al. (2023) repeat this for barometers in New Zealand, finding a comparable result. However, they also note that, at each station prior to the peak pressure, there is a gradual increase in pressure over a period of approximately 15 minutes. Thus, if you actually perform the linear regression using the times at which the Lamb wave onset is detected, rather than the peak, an origin time of 04:15 is retrieved. This is consistent with the timing of the primary seismoacoustic event which is recorded (Matoza et al., 2022). I therefore ask the authors if it would be more appropriate to use a Lamb wave origin time of 04:15. Alternatively, refer to the circle in Fig 2a as the location of the peak of the Lamb wave, rather than the leading edge.
5. Lines 320-324. The authors state that they are able to reproduce the observed particle radii oscillating in size with time. However, the evidence for this oscillation is not strong; based on the error bars shown in Fig 4a, the amplitude of this oscillation is smaller than the uncertainty on the measurements. Additionally, the modelled temporal oscillation does not appear to be in-phase with the observed oscillation (if it is there). The authors also state that the best agreement between modelled and observed results occurs for an assumed water vapour concentration of 5.45 ppmv. Can the authors quantify this level of agreement and provide an estimate of what range of concentrations still provide consistent results.
6. Fig. 4a. Why does the modelling shown here use 04:47 as an origin time? This is the time of the first Himawari image which observed the ice ring, so presumably ice crystal

growth must have started earlier than this? How sensitive is the result of 5.45 ppmv as the best fitting vapour concentration to the choice of origin time.

Minor comments

Scientific content

1. Line 44. Can you quantify the adiabatic decrease in ambient temperature?
2. Line 59. I suggest adding the word “explosive” in front of “volcanic eruptions”.
3. Lines 175-176. The sentence “Examples of such impulses ...” requires references. I’m particularly surprised that atmospheric Lamb waves have been observed to be generated by earthquakes.
4. Line 189. The text says that the Lamb wave caused a maximum temperature reduction of 8 K in the lower stratosphere. But this magnitude of decrease goes beyond the colour bar scale in Figure 2a, which only reaches -4 K. Surely the colour bar in Figure 2a should be extended to this.
5. Lines 190-193 and Figure 2b. The sentence “We estimate that these ...” and Figure 2b required me to do a bit of further reading to understand what is described here. I’m personally not familiar with weighting functions. However, from the context here and in the rest of the paper, I think they characterise the contribution to the signal in different wavelength bands from different parts of the Earth’s atmosphere. Some additional explanation and/or references might be needed for general readers who are not remote sensing scientists.
6. Lines 261-263. The authors state that the temperature perturbation caused by the Lamb wave at pressures of 100 hPa is predicted to be ~ 1.3 K and claim that this is sufficient to trigger ice nucleation. However, on line 249, the authors state that the perturbation needs to be at least 1-2 K. So, this inferred perturbation is only just large enough. It’s probably worth noting this.
7. Line 435. How has the value of 0.111 for the effective variance been chosen?

Presentation and style

1. Figure 1a. The white font makes it very difficult to read the label ‘300 km’.
2. Fig. 1. The caption needs to describe what the circle at 300 km corresponds to.
3. Throughout the manuscript, the vertical structure of the atmosphere is described in terms of pressure. Whilst this makes sense from an atmospheric perspective, it would help the more general reader if reference were also made to the altitudes being considered. For example, when referring to the pressure range 6-52 hPa, or to 100 hPa, it would be good to also state the corresponding altitudes.
4. Fig. 4. Please define AHI in the caption, as it isn’t defined in the text until the methods section. Also, the g parameter shown in the legend are missing units whilst the symbol γ is used in the caption but the symbol ζ is used in the text. It might also be useful to add a

reference to Equation 28 in the caption, to direct the reader to where the damped oscillator model is defined.

5. Line 414. Please state what the “effective radius” refers to. Is it ice crystals specifically, or can it be any particulate matter?
6. Line 426. Add the word “size” between “gamma” and “distribution”
7. Line 426. Add the word “index” between “refractive” and “given”
8. Line 527. Add the word “perturbation” between “temperature” and “time-series” to make it clear that the quantity given by equation 9 is the temperature perturbation, rather than the absolute temperature.
9. Line 541. Maybe add a reference to Fig. 3 after stating that “ $T_{\text{ambient}} = 193 \text{ K}$ at 100 hPa” to remind the reader where this value comes from.

References

Jarvis, P. A., Caldwell, T. G., Noble, C., Ogawa, Y. & Vagasky, C. (2023). Volcanic lightning reveals umbrella cloud dynamics of the January 2022 Hunga Tonga-Hunga Ha’apai eruption. *EarthArXiv*. <https://doi.org/10.31223/X5D09J>

Matoza, R. S., Fee, D., Assink, J., Iezzi, A. M., Green, D. N., Kim, K. et al. (2022). Atmospheric waves and global seismoacoustic observations of the January 2022 Hunga eruption, Tonga. *Science*, 377, 95-100. <https://doi.org/10.1126/science.abo7063>

Second review of

Transient ice ring observed during the 15 January 2022 eruption of Hunga volcano

Andrew T. Prata¹, Roy G. Grainger, Isabelle A. Taylor and Alyn Lambert

Previous comments are in **black**, responses in **blue** and new comments are in **green**

General:

The manuscript has improved significantly. The authors invested considerable effort into the study, and their work has paid off. The paper is now almost ready for publication. I only have a few comments/questions -both on previously raised issues and on some new points- which are listed below.

Specific comments:

- 1) line 120: 'The mean ice particle sizes were extremely small ($\sim 2 \mu\text{m}$; Fig 1d) when compared to tropical tropopause cirrus clouds [$2.5\text{--}25 \mu\text{m}$ radius; 20].'

The size of ice crystals depend on the stage of their development. All in-situ formed ice particles appear at very small sizes, because they form either heterogeneously on INPs or homogeneously on supercooled solution particles and thus have initially about the size of the particles, which is $< 1 \mu\text{m}$. In the course of the evolution of the ice cloud, they grow as long as the environment is supersaturated. The growth depends on the number of ice crystals, since the ice crystals compete for the available water vapor.

At the cold temperatures and low water vapor mixing ratios prevailing in the icing, the ice particles cannot grow to mean sizes substantially larger than $\sim 2 \mu\text{m}$, especially since the time for growth is short (see your line 120: 'the whole process (from formation to dissipation) took ~ 2 h to complete'). You show this in Figure 4 (the cooling / growth phase is only about 10min).

So the reason that in most cases the ice particles grow larger under natural TTL conditions is that the growth periods are longer, or that the ice crystal concentration is very small. A typical mean ice crystal size in the TTL is $\sim 8 \mu\text{m}$ (Krämer et al., 2020, ACP).

- I recommend to discuss in the paper that a shorter growth periods and/or a larger ice crystal concentration is likely the reason for the smaller ice particles observed in the ice ring in comparison to natural TTL ice clouds.

- Also, I recommend not saying that the ice crystals are extremely small, but that such small ice crystals are unlikely to occur in the atmosphere under natural conditions.

Thank you for these comments and suggestions. We appreciate these insights and have incorporated them into our revised manuscript. We couldn't find a statement or discussion in Krämer et al. (2020)

stating that “typical mean ice crystal size in the TTL is 8 μm ”. Looking at their figures (see Figure S2 for R_{ice}), it looks like the median at 15-18 km is colour-coded according to the 6-18 μm radius range.

The mean ice crystal size of $\sim 8 \mu\text{m}$ can be seen in Krämer et al. (2020), Figure 8, panel f, at the lowest temperatures ($<190\text{K}$).

We have revised the statement to make mention of both factors discussed (namely, ice crystal concentration and growth duration) and have deleted any mention of “extremely small” ice particles. The relevant revised text is: “The mean ice particle effective radii within the ice ring at 05:17~UTC was $\sim 2 \mu\text{m}$ (Fig. 1d). Tropical tropopause cirrus cloud sizes have been reported in the range from 2.5-25 μm radius (Woods et al., 2018); however, the size observed depends upon the stage of cloud development. The Hunga ice ring particle sizes would fall into the smallest, median R_{ice} size range (1-6 μm) reported by Krämer et al. (2016, 2020) who compiled a cirrus ice particle size climatology from in situ aircraft data. Krämer et al. (2020) show that the median ice particle sizes for cirrus clouds at similar altitudes ($\sim 15\text{-}18 \text{ km}$) lie in the range from 6-18 μm .” We have also added a footnote to explain how R_{ice} is defined.

2) line 123 ff: ‘.. ; however, the ice particles studied here are of an order of magnitude smaller, indicating a new ice nucleation pathway not previously documented for volcanic clouds.’

I think this statement is somewhat exaggerated - the observations can certainly be explained by conventional ice formation mechanisms - see the previous comment (on line 120).

In my opinion, the difference to previous volcano-induced ice clouds lies in the short period of the Lamb wave, and possibly in the number of ice crystals (see also previous comment). Is there any information about this? That would be helpful, and it would also be advisable to estimate the time scales for ice growth during previous volcanic eruptions.

Nevertheless, I find the observation and description of this phenomenon to be important.

We were unable to find published estimates of the timescales (e.g. in seconds or minutes) for ice particle growth in volcano-induced ice clouds. Durant et al. (2008), which we cite, provides a comprehensive overview of volcano-induced ice clouds. We agree with the reviewer that a possible reason for the small ice particle sizes is due to the short duration of ice supersaturation triggered by the atmospheric waves caused by the eruption. In response to this reviewer comment we have revised and added the following text:

“Ice particle sizes of $\sim 20 \mu\text{m}$ associated with volcanic eruptions have been observed before (Durant et al., 2008; Prata et al., 2020) and the particle sizes observed have been attributed to overseeding (i.e. high number concentrations of ice nucleating particles; Durant et al., 2008); however, the ice particles studied here are of an order of magnitude smaller, indicating an earlier stage of ice particle growth that has not previously been documented for ice clouds associated with volcanic eruptions. Such small ice particles are unlikely to be observed under natural conditions. In theory, the smallest ice particles will be observed at the very beginning of the ice particle growth phase during cloud development. Once nucleated, ice particles will initially grow via vapour deposition. Particle growth via vapour deposition is principally driven by the level of supersaturation, initial (dry) particle size distribution and, **in case of heterogeneous ice nucleation***, the ice nucleating particle (INP) number concentration (Mason, 1975; Rogers and Yau, 1989, Pruppacher and Klett, 2010; Lohmann et al., 2016). The longer an environment remains supersaturated with respect to ice, the larger an ice particle will be expected to grow. Thus if the environment is subject to only a short

period of supersaturation then small ice particles may be expected. For distributions of ice particles, low number concentrations will lead to larger particle sizes (and vice versa) as individual ice particles compete for the available background ambient water vapour.”

* this is important to mention because in homogeneous ice nucleation –the other pathway of ice formation- the number of ice particles does not depend on the number of homogeneously freezing aerosol particles.

I would also order the sentences by importance of the parameter: ,... driven by the level of supersaturation, and, in case of heterogeneous ice nucleation, the ice nucleating particle (INP) number concentration, and the initial (dry) particle size distribution.’

4) line 229ff: 'It is common in volcanic eruptions for water vapour in the atmosphere to be entrained into the plume, condense to liquid water and freeze in updrafts during transport to plume-top.'

Freezing of liquid drops is an additional way to form ice clouds in comparison to the two ways explained above (see e.g. Luebke et al., 2016, ACP). It is mostly triggered by INPs (thus heterogeneous), but liquid cloud drops would for sure be larger than 2 μm ! Plus, once frozen, the ice particles quickly grow to much large sizes. In case of high updrafts, liquid drops can survive until they reach the temperature of $\sim -40\text{C}$ and then freeze homogeneously - but even then the sizes would be around 10-20 μm . So I don't think that this pathway is possible here ...

Maybe this was the way the ice particles have formed in earlier volcanic eruptions ? (see also comment 3)

- I recommend to consider and explain in more detail the different ways of ice formation, as mentioned above.

To address both comments (3) and (4) we have revised Section 2.3 by:

1. Mentioning both pathways of ice formation (heterogeneous INPs and supercooled solution aerosol particles).
2. Mentioning the Rolf et al. (2012) study which shows how volcanic particles are efficient INPs.
3. Adding further discussion on ice cloud (cirrus) formation drawing on studies from Luebke et al. (2016), Krämer et al. (2016, 2020), Lawson et al. (2019) and Karcher et al. (2022). Our revised text is as follows:

“In order for ice clouds to form in the atmosphere there needs to be a source of liquid water or water vapour that is brought to supersaturation with respect to ice. For cirrus, recent work suggests that these clouds can be grouped into two broad categories: in situ origin cirrus and liquid origin cirrus (Luebke et al., 2016; Krämer et al., 2016, 2020). In situ origin cirrus clouds may form heterogeneously on ice nucleating particles (INPs) or may form homogeneously on supercooled solution aerosols (Kärcher et al., 2022 – there are much earlier fundamental references, e.g. Pruppacher and Klett, 1997). Liquid origin cirrus clouds typically contain larger ice particles than in situ origin cirrus as they form from liquid water drops which rapidly grow once frozen (Luebke et al., 2016; Lawson et al. 2019). Even in the case of strong updrafts, where liquid water droplets can survive until they reach $-38\text{ }^{\circ}\text{C}$ and freeze homogeneously, the smallest liquid origin cirrus ice particle sizes are typically $\sim 10\text{-}20\text{ }\mu\text{m}$ radius (Krämer et al., 2016).

Volcanic ash particles are known to be efficient INPs (Durant et al. 2008; Rolf et al. 2012) and it is common in volcanic eruptions for water vapour in the atmosphere to be entrained into the plume, condense to liquid water and freeze in updrafts during transport to plume-top (Woods et al. 1993; Tupper et al. 2009). Magmatic water, expelled during an eruption, is also a source of water for ice nucleation, meaning that volcanogenic ice can form in relatively dry atmospheres (Rose et al.

2004). A further source of water vapour can be added to the atmosphere via phreatomagmatic interactions (Prata et al. 2020). It is likely that all of the aforementioned mechanisms were at play during the Hunga eruption, which raises the question: Which ice nucleation pathway was responsible for the formation of the ice ring?”

New comment

1) line 316 ff of the revised version : *... we explored the hypothesis that the temperature perturbation generated by atmospheric waves in this altitude region triggered rapid, large-scale heterogeneous deposition nucleation in the UTLS.*‘

How sure are you that it was pure heterogeneous ice nucleation? It is not unlikely that ice formation initially begins heterogeneously, but then homogeneous freezing of supercooled solution aerosols sets in (see Rolf et al., 2012, their Figure 7). Perhaps both possibilities should be described.

New comment

2) line 427f of the revised version: *... four levels of ice supersaturation (0.1, 1, 5 and 10 %)*...‘

I wonder if the supersaturation is kept constant during the simulations or if it adjusts? That would be good to mention in the text.

New comment

3) line 500f of the revised version: *the presence of pre-existing ice clouds (and INPs) were necessary to explain the supersaturated conditions required to form the ice ring.*

Is this an assumption or a fact? Wouldn't the temperature drop caused by the wave have increased the supersaturation sufficiently to trigger ice formation even without previously evaporated clouds? According to Fig. 4, the temperature dropped by 5 degrees, which corresponds to an increase in the relative humidity above ice of approximately 50%. It would be great if this could be discussed more clearly.